# Endocrine therapy reprogramming of breast cancer facilitates metastatic escape via upregulation of P-Rex1/Rac1 signalling

Kristine J. Fernandez[1,17], Ghazal Sultani[1,2,17], Max Nobis[3], Brian Gloss[4], Leila Eshraghi[1,2], Amy E. McCart Reed[5], Sarah Alexandrou[1,2], Christine Lee[1], Daniel L. Roden[1,2], Emily I. Jones[6], Maryam Hasha Simad[1,2], Ewan K. A. Millar[7,8], Nenad Bartonicek[1], Samantha R. Oakes[2,9,10], Fatima Valdes-Mora[2,11], Yolanda Colino-Sanguino[11], Ellie T. Y. Mok[1,2], Hannah L. Williams[1,12], Jamie R. Kutasovic[5], Margaret C. Cummings[5,13], Janett Stoehr[1], Victoria Lee[1], Kate Harvey[1], Sunny Wu[1,2], Sunil R. Lakhani[5], Peter T. Simpson[5], Thomas R. Cox[1,2], Lisa M. Ooms[6], Christina A. Mitchell[6], Rob Salomon[14], Alexander Swarbrick[1,2], David Gallego-Ortega[1,2,15], Elgene Lim[1,2,16], Paul Timpson[1,2] & C. Elizabeth Caldon[1,2] ✉

The estrogen receptor (ER) drives growth in most breast cancers. Endocrine therapy reduces recurrence, however around 30% of cancers relapse. Many recurrences occur years later, with slowly proliferating, hard-to-treat disease. To study this, we generate slow-growing resistant cells that form small primary tumours but readily metastasise. Single-cell RNA sequencing (scRNAseq) reveals that endocrine therapy reprograms these cells, notably upregulating the Rac1 signalling component P-Rex1. We find in clinical cohorts that P-Rex1 is high in ER+ breast cancer, including in late recurrent disease. Intravital imaging demonstrates that Rac1 signalling is active in ER+ cells following endocrine therapy. Targeting the Rac1 pathway with small molecule inhibitors (NSC23766, R-ketorolac) reduces survival and motility in resistant cells, inhibits in vivo Rac1 activity, and reduces tumour burden when combined with tamoxifen in a drug-refractory patient derived xenograft model. This work identifies the P-Rex1/Rac1 axis as a potential therapeutic target for late recurring ER+ breast cancer.

Endocrine therapy (e.g. tamoxifen, fulvestrant, aromatase inhibitors) is the backbone of adjuvant therapy for the ~75% of breast cancer patients who have ER+ breast cancers. After initial surgery and treatment, patients receive endocrine therapy for 5–10 years to inhibit the growth of residual ER+ cancer cells. However, up to 30% of these patients develop incurable endocrine therapy resistant recurrences[1]. Commercially available gene signature tests (e.g. Oncotype DX) predict early and highly proliferative recurrences[2] and provide a therapeutic decision tool on the addition of adjuvant chemotherapy to

enhance treatment[3]. However, ~50% of recurrences occur after >5–10 years of endocrine therapy[3,4], and lead to ~50% of recurrence-related death. Unlike early recurrences, late recurrences lack a clear prognostic gene signature, and do not include the proliferative biomarkers that are associated with early recurrence. Late recurring ER+ breast cancers are an area of unmet need due to high mortality, difficulties in predicting the likelihood of occurrence, and lack of specific treatment.

The current dogma is that late ER+ breast cancer recurrences are due to the reactivation of dormant cancer cells[5,6]. Dormancy can

manifest as "cellular dormancy" where cells enter a state of prolonged cell cycle arrest, or "tumour mass dormancy" where the proliferation of slowly cycling cells is counteracted by cellular turnover, and metastases do not reach the threshold of detection. Significant efforts have been made towards exploring mechanisms of cellular dormancy, but the slow proliferative rate of primary ER+ cancer cells puts weight to the argument that tumour mass dormancy is an alternative route to late recurrence. In particular, measurements of breast cancer cell proliferation in tumours shows a large range of doubling time (10–700 days), and cancers with doubling times of >250 days are estimated to take >15 years to become detectable[7,8].

To investigate the role of slow growing cells in ER+ breast cancer recurrence, we initially isolated and characterised slowly growing endocrine therapy treated cells compared to their rapidly growing counterparts. We identified that these slow growing cells emerge immediately after the administration of endocrine therapy, and they have metastatic potential despite a slow proliferative rate. Investigation of the transcriptome of these cells by scRNAseq revealed an upregulation of Rac1 signalling, which we then validated as a signature of poor response across patient cohorts and as a targetable axis in the MMTV-PyMT model and a PDX model of breast cancer.

## Results

### ER+ metastases frequently have lower proliferation than their matched primary tumours

Little is known about the relative rate of metastatic growth in metastatic lesions of ER+ breast cancer compared to their matched primary tumours. Using matched primary and metastatic tissues, we investigated the relative rate of proliferation of ER+ metastases compared to matched primary tumours, using Ki67 immunohistochemistry as a surrogate for proliferation. We examined an autopsy cohort of breast cancer patients[9] with a median time between diagnosis and autopsy of 24 months. Primary cancers from ER+ breast cancer patients with multiple matched metastases showed less Ki67 staining in metastatic lesions, whereas there was no significant change in Ki67 levels between primary and metastatic disease in ER- breast cancer (Fig. 1A, B). Overall, ER+ metastases can have lower proliferative rates than their matched primary cancers.

### Development of slow growing endocrine therapy resistance models

Often models of drug resistant cancer cells are derived by selection of the most rapidly proliferating clones, but the prevalence of slow proliferating ER+ metastatic disease led us to specifically isolate slow growing populations. We observed that MCF-7 cells treated with endocrine therapy (tamoxifen, fulvestrant, estrogen deprivation) emerge with proliferative heterogeneity after 1–2 months, including colonies of 2–3 cells and colonies of >100 cells (Fig. 1C). To isolate slow growing cells we treated MCF-7 ER+ breast cancer cells with fulvestrant for 20 days and then labelled with a live cell intracellular dye that is diluted with each cell division (CellTrace). Cells were cultured for a further 12 days to distinguish cells with different levels of CellTrace (Fig. 1D), and three subpopulations separated by flow cytometry (Fig. 1D). Of note, the sorted cell populations were all actively proliferating and had undergone at least 1 cell division with each CellTrace labelled cell sort (Supplementary Fig. S1A).

Re-cultured cells retained the ability to grow at different rates in fulvestrant (Fig. 1E). We noted that the slow growing population developed the occasional larger colony with continuous passaging, and in order to maintain a low proliferative capacity, we performed sequential cell sorting and selected slow growing cells over 21 months by two further rounds of CellTrace labelling. We designated the resulting slow growing, but resistant, population "Endocrine Tolerant". We also maintained the fast-growing therapy resistant cell line isolated in Fig. 1D, which we designated "Fast".

We examined the relative ability of cells to grow in the presence of 17β-estradiol, fulvestrant or tamoxifen (Supplementary Fig. S1B–D). All cell lines grew in media supplemented with 17β-estradiol, but the Endocrine Tolerant cells were intrinsically less proliferative (Supplementary Fig. S1B) despite expressing similar levels of ER to Fast cells (Supplementary Fig. S1E, F). Both Endocrine Tolerant and Fast cells appeared resistant to fulvestrant and tamoxifen, but Fast cells proliferated more rapidly in the presence of either drug (Supplementary Fig. S1C, D). Overall, we had isolated two unique resistant breast cancer derivatives with different proliferative capacities.

### Xenografts of Endocrine Tolerant cells led to smaller primary tumours with enhanced collagen deposition, but metastasised with the same efficiency

We next assessed if the slow proliferative capacity of Endocrine Tolerant cells also manifested in vivo. We performed intraductal xenografts[10], where cells are injected into the mammary gland via the nipple, leading to high fidelity tumour development within the mammary duct and metastasis to the lung and bone, among other sites, as is typical of ER+ breast cancer (Fig. 1F). We performed intraductal xenografts of Endocrine Tolerant and Fast cells, and monitored primary and secondary tumour development over 9 months.

60% (6/10) of the Endocrine Tolerant xenografts developed palpable primary tumours, whereas 100% (10/10) of Fast-growing resistant xenografts developed palpable tumours following the injection of equal numbers of cells (Fig. 1G, H). Fast-growing tumours at endpoint (ethical endpoint, or nine months) were significantly larger than Endocrine Tolerant tumours ($p < 0.0021$) (Fig. 1H).

Interestingly, primary tumours of Endocrine Tolerant cells appeared morphologically distinct from those derived from Fast cells (Fig. 1I). Endocrine Tolerant tumours had significantly higher deposition of highly bundled and aligned collagen than Fast-growing tumours, as measured by multi-photon based second harmonic generation (SHG) microscopy (Fig. 1J), which is a feature associated with cellular senescence[11]. Individual Endocrine Tolerant cells within tumours also had significantly larger nuclei that appeared distended, or were multi-nucleated (Fig. 1K/L), which is also reminiscent of senescent-like cells[12]. We did not observe any significant numerical differences in cancer-associated fibroblasts between tumours, as measured by α-SMA immunohistochemistry (Supplementary Fig. S1G/H).

While there were different rates of primary tumour formation from Endocrine Tolerant and Fast-growing cells, these differences were less apparent in metastases. Lung metastases occurred in all Endocrine Tolerant xenografted animals (10/10), regardless of whether the primary tumour was palpable, and the size of these metastases was not significantly different to Fast-growing xenografts (Fig. 1M). Bone metastases were detectable in a subset of both Fast-growing (40%, 2/5) and Endocrine Tolerant xenograft (50%, 1/2) models (Fig. 1N, Supplementary Fig. S1I). 8 of 10 mice with Endocrine Tolerant xenografts showed evidence of gynaecological abnormalities including ovarian hyperplasia, haemorrhagic ovarian cysts and uterine hyperplasia, whereas this occurred in 4 of 9 mice examined with Fast-growing xenografts (Fig. 1O, Supplementary Fig. S1J).

Overall, our unique model demonstrates that slow growing Endocrine Tolerant cells form small primary tumours with large nuclei and enhanced collagen deposition, but these cells are capable of metastasising to multiple sites.

### Endocrine Tolerant cells are transcriptionally distinct to Fast growing and parental cells

As shown in Fig. 1C, different proliferative rates of cells had manifested as different colony sizes by 28 days following drug selection, and this proliferative potential is maintained in vivo in primary tumours. This implies that cells diverge with different anti-proliferative responses

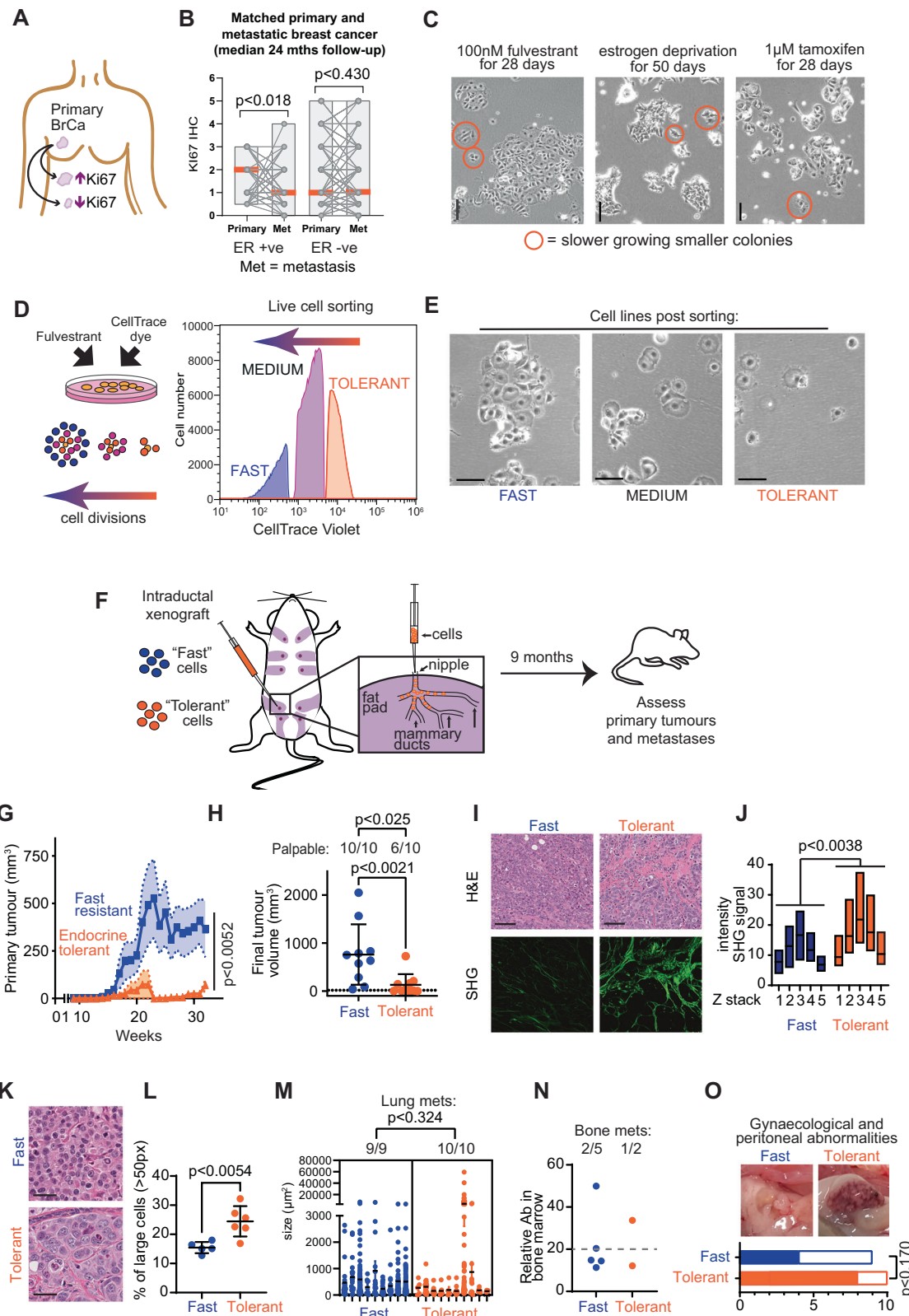

shortly after treatment and this is sustained over time. From this, we hypothesised that Endocrine Tolerant cells will be a subpopulation with altered proliferative potential that either (a) develops from persister cells within the original parental population or (b) arises during initial selection with endocrine therapy.

We investigated these possibilities by performing scRNAseq of our resistant cell populations and compared these to parental MCF-7

cells. Endocrine Tolerant cells did not bear resemblance to one particular subgroup of parental cells (Fig. 2A), in line with a prior study on a model of ER+ aromatase therapy resistance which suggests that endocrine therapy resistance does not arise from particular subclones[13]. This does not eliminate the possibility of a persister cell population as we may not have examined sufficient cells, or used the most sensitive method to detect persister cells. However, without a

**Fig. 1 | Proliferative diversity following endocrine therapy is sustained long-term in tumours. A** Schematic of breast cancer (BrCa) metastases with various degrees of proliferation. **B** Matched primary and metastatic ER+ ($n = 18$ primary, $n = 95$ metastases) and ER- cancers ($n = 38$ primary, $n = 163$ metastases) were compared for relative Ki67 by IHC. Grey lines indicate the change in Ki67 between primary and matched metastasis. Box plot, bounds=range, orange bar = median. Analysed by paired two-sided t-tests. **C** Images of proliferative heterogeneity in MCF-7 cells treated with 100 nM fulvestrant for 28 days, deprived of estrogen for 50 days, or treated with 1 μM tamoxifen for 28 days. Representative images of $n = 3$ experiments. Slow growing colonies indicated by orange circles; scale bar = 100 μm. **D** MCF-7 cells cultured for 20 days with 100 nM fulvestrant were labelled with CellTrace for 12 days before live cell sorting into three populations with different proliferative rates. **E** Phase contrast images of fulvestrant resistant cells sorted into Fast, Medium and Tolerant cells after 5 weeks of culture, representative of >3 independent images. Scale bar = 100 μm. **F** Schematic of intraductal technique. **G** Primary tumour growth of Fast and Endocrine Tolerant tumours ($n = 10$ each). Analysed by two-way ANOVA mixed effect model. Error bars are SEM. **H** Endpoint tumour volume ($n = 10$ each) of Fast and Endocrine Tolerant cells, analysed by two-

sided Mann–Whitney test. Palpable tumours analysed by Chi² test. Dashed line indicates the threshold for palpable tumours. Error bars are SEM. **I** Representative image of H&E staining and collagen detection by second harmonic generation (SHG) signal in Fast and Endocrine Tolerant tumours. Scale bar = 100 μm. **J** Quantification of peak SHG signal intensity in Z stacks (6 tumours per condition). Floating bars show minimum to maximum with line at mean, and analysed by two-way ANOVA. **K** Representative H&E section of Fast and Endocrine Tolerant tumours to detect nuclei size. Scale bar = 40 μm. **L** Quantification of cells with large nuclei (>50 pixels) from Fast tumours ($n = 5$) and Endocrine Tolerant tumours ($n = 6$). Analysed by two-tailed t-test, error bars are SEM. **M** Quantification of lung metastases from individual mice ($n = 9$ Fast, $n = 10$ Endocrine Tolerant) by cytokeratin IHC, analysed by two-sided nested t-test. Error bars are SEM. **N** Bone metastases assessed by flow cytometry from bone marrow. Ab = human CD298 antibody. Each dot represents a different animal. Dashed line indicates > background staining. **O** Gynaecological abnormalities associated with xenografts ($n = 9$ Fast, $n = 10$ Endocrine Tolerant), analysed by Chi² test. Source data are provided as a Source Data file.

strong indication of persister cells, we hypothesised that differences in proliferation during resistance could arise from heterogeneity between individual cells in their response to endocrine therapy.

In order to identify heterogeneity in the response of cells to fulvestrant, we performed scRNAseq on MCF-7 cells undergoing arrest with fulvestrant and subsequently re-entering the cell cycle following estrogen stimulation. We analysed untreated MCF-7 cells, cells arrested with 10 nM fulvestrant for 48 hours, and fulvestrant-arrested cells stimulated with estrogen for 6 hours or 24 hours (Fig. 2B). We confirmed that these treatments induced regulation of cell cycle proteins consistent with cell cycle arrest by fulvestrant and cell cycle re-entry following estrogen treatment (Fig. 2C). We then annotated cells with a transcriptional cell cycle signature. We found that a transcriptional S phase signature was activated within 6 h of estrogen stimulation (Supplementary Fig. S2A), consistent with previous observations of the early induction of S phase genes following estrogen stimulation of fulvestrant arrested MCF-7 cells[14,15]. Cells then demonstrated accumulation in S and $G_2$/M phase after 24 h of estrogen treatment using propidium iodide staining (Supplementary Fig. S2B), consistent with previous studies[14,15].

We next examined how fulvestrant-arrested and estrogen-stimulated cells transition through different states by plotting cells on a transcriptional trajectory using pseudotime analysis[16]. This identified cells transitioning between six different nodes (Fig. 2D). Estrogen treatment for 6 h superimposed with four of the nodes, and these were enriched with S phase transcriptional signatures. Estrogen treatment for 24 h localised across several nodes, but notably co-occurred with an area of $G_2$/M transcriptional signature enrichment that was situated between nodes (Fig. 2E). Fulvestrant treatment associated predominantly with $G_1$ transcriptional signatures, which co-localised with the remaining two nodes (labelled Node #1 and Node #2). Node #1 and Node #2 were notably distant from one another on the x_pseudo axis (Fig. 2D/E), which suggested that the two nodes enriched for fulvestrant treatment could be distinct in terms of their transcriptional programming. We observed that the two resistant cell lines, Fast and Endocrine Tolerant, were co-located with Node #2 of fulvestrant treatment (Fig. 2D/E), so we investigated the transcriptional signatures of the fulvestrant treatment enriched nodes in more detail to understand their differences.

We compared the transcriptomes of the two fulvestrant enriched nodes (Fulvestrant Node #1 and Fulvestrant Node #2) to each other, and to the remainder of the cells in the analysis. Gene set enrichment analysis (GSEA) showed that there was enrichment of different pathways in the two fulvestrant node populations (Fig. 2F). Node #1 population had a $G_2$/M signature indicative of mitotic exit. By contrast, Node #2 showed overall downregulation of $G_2$/M transition as well as

persistent signatures of respiratory electron transport, which can be associated with perturbed mitochondrial function in cellular senescence[17] (Fig. 2F). We identified that *HMGB2*, whose reduction is a key indicator of senescence[18], is lower in Node #2 cells compared to other populations (−2.10 logFC, Fig. 2G). This indicated that Node #2 could be a pseudo-senescent population as senescent cells frequently skip mitosis upon cell cycle exit[19], have mitochondria-associated respiratory electron transport[20], and have reduced *HMGB2*[18].

We next performed clustering analysis of the cells within the fulvestrant enriched Node #2 to determine the similarities of Endocrine Tolerant cells and Fast-growing cells to this subset of fulvestrant arrest. We found that Endocrine Tolerant cells clustered more closely with fulvestrant-arrested cells than the Fast growing population (Fig. 2H). Heatmap analysis of the most significantly altered genes showed that genes that were enriched in Endocrine Tolerant cells were also altered in a proportion of the cells arrested with fulvestrant for 48 hours (Fig. 2I). Top genes upregulated in both Endocrine Tolerant and fulvestrant arrested cells (*PREX1*, *CPE*, *NPNT*, *KY*, *MALAT1*) included several associated with metastatic and growth promoting functions in cancer. *PREX1* is a component of the Rac1 signalling pathway that is associated with breast cancer metastasis[21], *CPE* suppresses apoptosis to promote tumour growth and metastasis in multiple cancer types[22], *NPNT* encodes nephronectin, an extracellular matrix protein that enhances bone metastasis[23], and the lncRNA *MALAT1* has been associated with both the activation and suppression of breast cancer metastasis[24,25].

Since the Node #2 population was highly enriched for senescence-like pathways, we stained fulvestrant treated, and Endocrine Tolerant and Fast resistant cells for senescence-associated β-galactosidase. β-galactosidase staining was increased with fulvestrant treatment, and it was sustained in the Endocrine Tolerant population, and there was a trend for Endocrine Tolerant cells to have higher β-galactosidase than Fast-growing cells (Fig. 2J/K). Endocrine Tolerant cells also generally had a larger more flattened morphology in culture (Supplementary Fig. S2C). This was also consistent with the larger cell size of Endocrine Tolerant cells that we observed in vivo (Fig. 1K/L).

Overall, it appears that Endocrine Tolerant cells are similar to a subpopulation of fulvestrant treated cells that go into a senescent-like arrest. This suggests that it is possible that Endocrine Tolerant cells are derived from a subpopulation of slow growing "senescent-like" cells that are programmed during endocrine therapy treatment.

## Divergence of signalling in resistance in association with Fast-growing or Endocrine Tolerant populations

We next compared which pathways were activated in Endocrine Tolerant versus Fast-growing cells and parental cells based upon the most

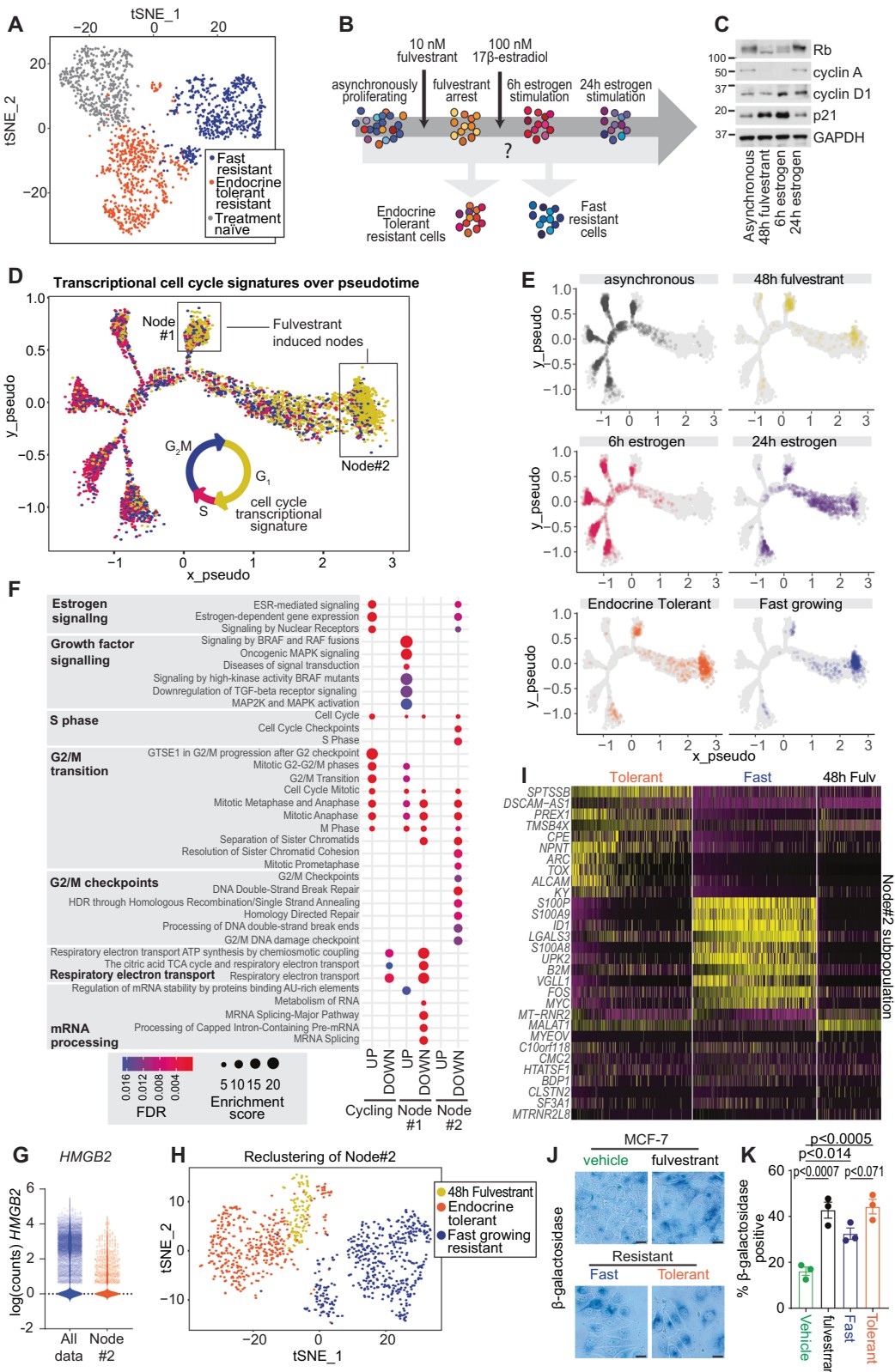

significantly altered genes from scRNAseq analysis. GSEA showed that Endocrine Tolerant cells had highly enriched signatures for luminal type breast cancers and endocrine therapy resistance (Fig. 3A). By contrast, Fast-growing cells had enrichment for genesets involving the upregulation of growth factor receptor signalling pathways, including EGF and neuregulin signalling (Fig. 3A). We additionally compared the Endocrine Tolerant cells to genes enriched in a dormancy model of ER

+ breast cancer[26]. This did not identify a large overlap of genes, and a similar proportion of overlap was seen between Fast-growing cells and the dormancy gene set (Supplementary Fig. S3A/B).

We next investigated gene regulatory networks within significantly altered genes of Endocrine Tolerant and Fast-growing resistant cells, followed by Steiner Forest network analysis to identify high-confidence networks[27]. Network analysis of both resistance

**Fig. 2 | scRNAseq to identify cell cycle transcriptional states induced by endocrine therapy, and overlap with Endocrine Tolerant cells. A** Multidimensional reduction analysis of principal components from parental MCF-7 cells ($n = 517$), Endocrine Tolerant ($n = 659$) and Fast-growing ($n = 578$) cells analysed using DropSeq. **B** Schematic of treatment to examine short-term fulvestrant-mediated arrest and release. MCF-7 cells were treated with 10 nM fulvestrant for 48 hours to induce a cell cycle arrest. Cells were then stimulated to re-enter the cell cycle with 100 nM 17ß-estradiol for 6 hours and 24 hours. **C** Western blot ($n = 2$) of total Rb, cyclin A, cyclin D1, p21 and GAPDH expression in short-term fulvestrant and estrogen treated MCF-7 cells. The samples derive from the same experiment, but one gel for cyclin A, cyclin D1, p21 and GAPDH, and another gel for Rb, were processed in parallel. **D** Pseudotime trajectory of scRNAseq data from asynchronous MCF-7 ($n = 2336$), 48 h fulvestrant treatment ($n = 340$), 6 h ($n = 2066$) and 24 h ($n = 941$) estrogen treatment of MCF-7 cells and Endocrine Tolerant ($n = 659$) and Fast-growing ($n = 578$) cells. Pseudotime trajectory constructed using Monocle 2. Transcriptional cell cycle signatures for $G_1$ (mustard yellow), S phase (pink) and $G_2$/M (blue) were superimposed upon the trajectory. Two fulvestrant arrested populations enriched for $G_1$ are indicated with the boxes "Node #1" and "Node #2". **E** Pseudotime trajectory of scRNAseq of separated asynchronous ($n = 2336$), 48 h fulvestrant ($n = 340$), 6 h estrogen ($n = 2066$) and 24 h ($n = 941$) estrogen treatment groups, as well as Endocrine Tolerant ($n = 659$) and Fast-growing resistant ($n = 578$) cells. **F** Gene set enrichment analysis of upregulated and downregulated genes in Node #1 and Node #2 cells compared to the remainder of cycling cells. **G** *HMGB2* expression in Node #2 ($n = 994$) compared to the entire cell population ($n = 6920$). **H** Multidimensional reduction analysis of principal components from cells in Node #2 from the pseudotime trajectory. Includes the Node #2 populations of short-term fulvestrant-treated MCF-7 cells ($n = 115$, mustard yellow), Endocrine Tolerant ($n = 401$, orange) and Fast-growing cells ($n = 478$, blue). **I** Heatmap of gene expression of top differentially regulated genes in Node #2 subpopulation. Yellow = high expression and purple = low expression. **J** Brightfield images of MCF-7 cells treated for 48 hours with vehicle (ethanol) or 100 nM fulvestrant, and Fast-growing and Endocrine Tolerant cells stained with senescence-associated ß-galactosidase. Scale bar = 20 μm. Representative of three biological replicates. **K** Quantification of β-galactosidase positive cells. Data ($n = 3$ biological replicates) analysed by one-way ANOVA with Tukey's multiple comparison test. Error bars are SEM. Source data are provided as a Source Data file.

models identified a major node of *JUN*, which is a component of the AP-1 complex that is activated downstream of growth factor receptors in tamoxifen resistance[28] (Fig. 3B/C). Also present in both was a *HES1* node, which is known to coincide with cells situated for longer in $G_1$ of the cell cycle[29], and notably this node is more prominent in the Endocrine Tolerant cells[29]. Fast growing cells had unique nodes of epidermal growth factor receptor (*EGFR*), consistent with upregulated growth factor receptor signalling; and *MYC*, a known mediator of anti-estrogen resistance[30]. *RAC1*, which coordinates cytoskeletal organisation and cell growth, was the highest confident unique gene signalling node identified from the transcriptome of the Endocrine Tolerant cells (Fig. 3C).

We validated the activation of different signalling pathways in Endocrine Tolerant and Fast-growing cells using MAGPIX protein arrays and western blots. Fast-growing resistant cells showed higher basal levels of EGFR in growth factor depleted media compared to Endocrine Tolerant cells (Fig. 3D/E), and pEGFR showed greater induction by serum and estrogen in Fast-growing cells. The addition of EGF or serum to starved cells led to a significantly greater upregulation of pHER2 in Fast-growing cells compared to Endocrine Tolerant cells (Fig. 3F/G).

The *RAC1* signalling node of Endocrine Tolerant cells included central genes *RAC1* and *PREX1*. The Rac1 signalling pathway drives invasion, cell survival and metabolism, and Rac1 pathway genes *RAC1* and *PAK2* have previously been implicated in endocrine therapy resistance[31,32]. *PREX1* encodes the Rac1 guanine nucleotide exchange factor, P-Rex1, which has pro-migratory roles in metastasis[33–35] and is implicated in metastasis of ER+ cancer. *PREX1* was upregulated in Endocrine Tolerant cells that overlapped with senescent-like arrest induced by 48 hours of fulvestrant treatment (Fig. 2I, Fig. 3H). We confirmed that P-Rex1 protein was expressed at higher levels in Endocrine Tolerant cells than Fast-growing cells, although no increase was seen in Pak2 or Rac1 (Fig. 3I/J). We also assessed short-term regulation by anti-estrogen and estrogen treatment in parental cells, and P-Rex1 was slightly, but significantly, upregulated by 48 hours of fulvestrant treatment, and then further upregulated by the addition of estrogen (Fig. 3K/L).

P-Rex1 in ER+ cells has been associated with downstream activation of insulin growth factor receptor (IGFR) signalling, insulin receptor (IR) signalling and extracellular signal-regulated kinase (ERK) signalling[36,37]. We found that the Endocrine Tolerant cells had significantly enhanced IGFR and IR signalling compared to Fast cells (Fig. 3M/N/O), in line with previous reports. It has also been reported that P-Rex1 enhances ERK signalling in some model systems[37,38], but not others[39], and we did not observe significant changes in ERK signalling (Supplementary Fig. S4A/B).

To determine whether P-Rex1 was important to the oncogenic potential of Tolerant cells, we performed *PREX1* shRNA experiments and examined the effect on the migratory ability of cells, which is an important facet of P-Rex1 activity in cancer[33]. *PREX1* shRNA reduced expression of P-Rex1 in both the Endocrine Tolerant and Fast-growing resistant cells (Fig. 3P). Endocrine Tolerant cells expressing *PREX1* shRNA showed a ~ 37% impairment in the migration of cells compared to a non-targeting control, whereas *PREX1* shRNA derivatives of Fast-growing cells did not show a significant difference in migration potential compared to control (Fig. 3Q). Thus, in Endocrine Tolerant cells the expression of P-Rex1 is important for migration.

Since P-Rex1 and EGFR upregulation did not co-occur in the Endocrine Tolerant and Fast-growing cells, we investigated whether this mutual exclusivity was evident in patients. We found that *EGFR* and *PREX1* are negatively correlated in primary ER+ cancers (TCGA: Fig. 3R, Metabric: Supplementary Fig. S4C) and metastatic ER+ cancers ($n = 67$, The Metastatic Breast Cancer Project[40], Fig. 3S), suggesting that EGFR and Rac1 signalling are activated in different subsets of ER+ disease.

### Elevated P-Rex1 expression in the MMTV-PyMT model of mammary cancer

Next, we assessed whether Rac1 activity was induced in other models following treatment with endocrine therapy. We chose the MMTV-PyMT mouse model of breast cancer as it recapitulates key aspects of the histopathology, disease progression and metastatic profile of luminal type breast cancer including ER+ hyperplasia (5 weeks), early carcinoma (8–12 weeks) and late carcinoma with metastatic disease to the lung at 10-14 weeks of age[41]. We have previously shown that Rac1 signalling is spatially concentrated to the invasive edge and vasculature of primary tumours in this model, and it is active in cells metastasising to the lung[42]. Analysis of microarray data (GSE43566[43]) from a transplant model of MMTV-PyMT mice shows that expression of *Rac1*, *Prex1* and *Pak2* are low in adenoma and carcinoma, but significantly increase during cancer cell dissemination in this model (Fig. 4A).

We then determined how Rac1 activity becomes altered in this model in response to endocrine therapy, using a Rac1-FRET biosensor[44]. This biosensor allows Rac1 activity to be monitored in real-time with fluorescent lifetime imaging microscopy (FLIM) by measuring the lifetime of a donor fluorophore with the Rac1-Raichu FRET reporter, including longitudinal assessment of Rac1 activity in vitro and in vivo following drug treatment[42]. As previously described, a Rac1-FRET biosensor mouse was crossed to the MMTV-PyMT mouse model to create the MMTV-PyMT Rac1-FRET mouse model[42]. Tumours were allowed to develop for 12-18 weeks, and primary mammary tumours were excised and dissociated to generate MMTV-PyMT Rac1-

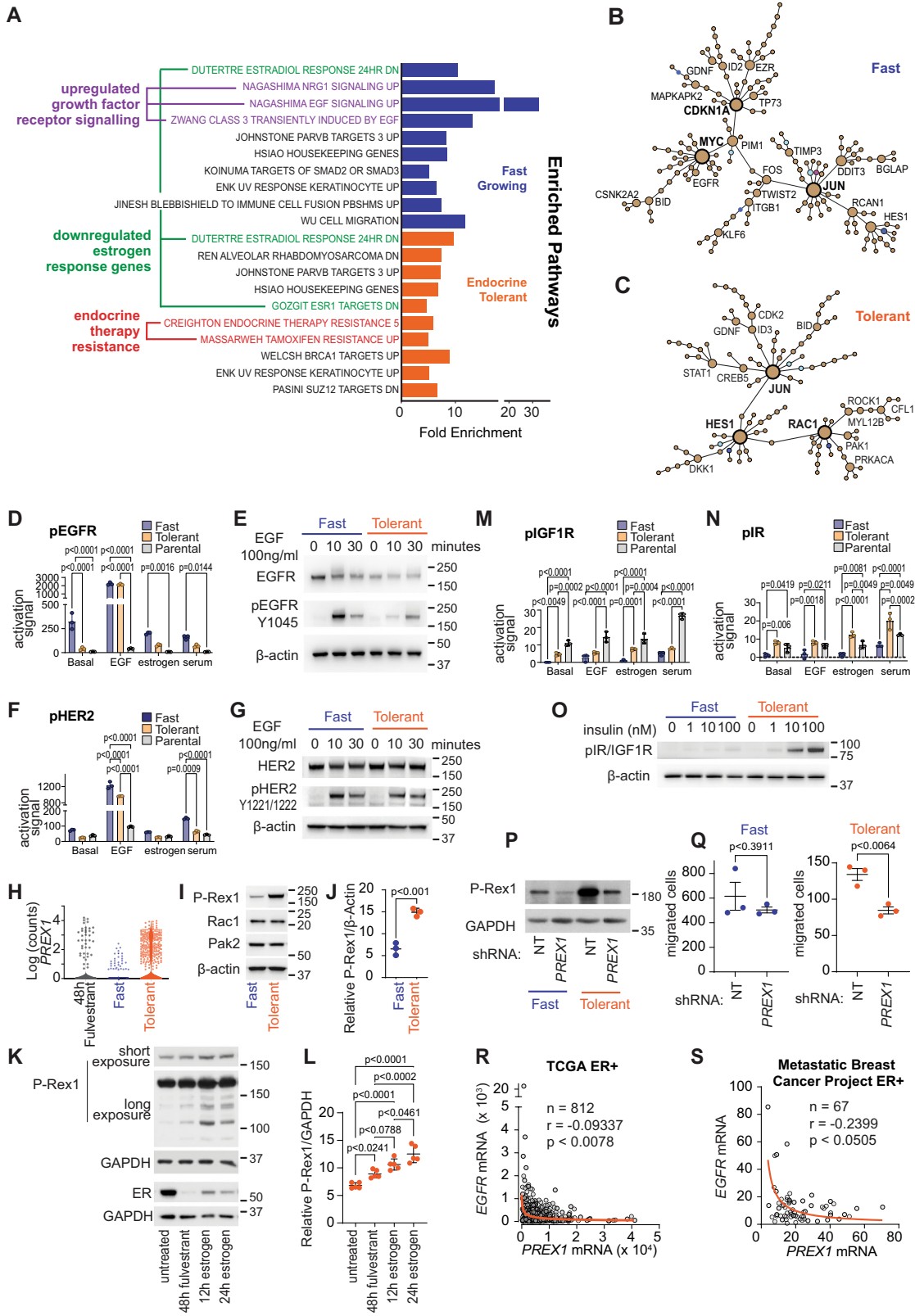

FRET cell lines (Fig. 4B). As the MMTV-PyMT model can lose ER expression upon disease progression[45], we chose a cell line with expression of ER (Fig. 4C). MMTV-PyMT Rac1-FRET cells were chronically exposed to tamoxifen for ~30 passages until reaching a constant proliferative rate (PyMT-Tam cells). Treated cells initially showed significant heterogeneity of colony size, indicating the presence of fast and slow growing colonies (Supplementary Fig. S5A). However, by the

end of selection PyMT-Tam cells grew significantly more slowly than the parental cell line (23.2 h vs 35.8 h doubling time, p < 0.0001, Supplementary Fig. S5B). Western blots showed that there was elevated P-Rex1 and Pak2, but not Rac1, following chronic tamoxifen exposure (Fig. 4C/D, Supplementary Fig. S5C).

We assessed Rac1 activity in tamoxifen exposed (PyMT-Tam) cells compared to parental cells and observed active Rac1 (high-FRET,

**Fig. 3 | Divergent signalling in Fast-growing and Endocrine Tolerant cells. A** Gene set enrichment analysis on Fast-growing compared to parental cells (n = 288 genes) and Endocrine Tolerant resistant cells compared to parental cells (n = 288 genes) using Curated.MSigDB. **B/C** Steiner Forest signalling network analysis in upregulated genes (n = 288) of **B** Fast-growing and **C** Endocrine Tolerant cells. Node size indicates number of attached edges, with nodes with >8 in bold. Node colours: Brown=genes, light blue=phenotype, dark blue=protein complex, pink=protein family. **D** Multiplexed analysis of EGFR phosphorylation in: basal cells, 30 min 100 ng/mL EGF, 30 min 100 nM 17 β-estradiol, or 30 min serum stimulation. Analysed by two-way ANOVA with Bonferroni's multiple comparison test from n = 3 biological replicates, error bars=SD. **E** Western blot of EGFR, pEGFR (Y1045) and β-actin in Fast and Endocrine Tolerant cells after 100 ng/mL EGF treatment for indicated times, representative of 3 biological replicates. **F** Multiplexed analysis of HER2 phosphorylation following treatment described in **D**. Analysed by two-way ANOVA with Bonferroni's multiple comparison test. **G** Western blot of HER2, pHER2 (Y1221/1222) and β-actin in Fast and Endocrine Tolerant cells after 100 ng/mL EGF stimulation for indicated times, representative of 3 biological replicates. (**H**) *PREX1* transcripts from scRNAseq analysis of MCF-7 cells treated for 48 hours with fulvestrant (n = 340 cells), and in Fast-growing (n = 578 cells) and Endocrine Tolerant (n = 659 cells). **I** Western blot of P-Rex1, Rac1, Pak2 and β-actin in Fast and Endocrine Tolerant cells, representative of 3 biological replicates. **J** Densitometry of P-Rex1 normalised to β-actin from n = 3 biological replicates. Analysed by two-sided t-test, error bars=SD. **K** Western blot of P-Rex1, ER and GAPDH in short-term fulvestrant and estrogen treated MCF-7 cells. Representative of 5 replicates. **L** Densitometry of P-Rex1 normalised to β-actin from n = 5 biological replicates, analysed by one-way ANOVA with Tukey's multiple comparisons, error bars=SEM. **M/N** Multiplexed analysis of **M** pIGF1R and **N** pIR phosphorylation following treatment described in **D**. Analysed by two-way ANOVA with Bonferroni's multiple comparisons. **O** Western blot of pIR/pIGF1R and β-actin in Fast and Endocrine Tolerant cells after 30 mins insulin. Representative of 3 replicates. **P** Western blot for P-Rex1 and GAPDH in Fast and Endocrine Tolerant cells expressing non-targeting (NT) and *PREX1* shRNA, representative of n = 2 biological replicates. **Q** Quantitation of transwell migration assay of Fast and Endocrine Tolerant cells that express NT and *PREX1* shRNA. Analysed by unpaired two-sided t-test from n = 3 biological replicates, error bars=SEM. **R/S** Correlation between *PREX1* and *EGFR* in the (R) TCGA (RNAseq RSEM; n = 812 cancers) and **S** Metastatic Breast Cancer Project (RNAseq RSEM; n = 67 cancers). r=Spearman's correlation. Orange line is least squares fit. Source data provided in Source Data file.

represented by green/blue spectrum) (Fig. 4E). Quantitation of the fluorescent lifetime of the donor fluorophore, enhanced cyan fluorescent protein (ECFP), demonstrated a significant decrease in signal lifetime following tamoxifen treatment, indicating an increase in Rac1 activity (Fig. 4F).

Having established that Rac1 activity increases following chronic tamoxifen treatment of ER + MMTV-PyMT cells, we next assessed Rac1 activity in the context of chronic tamoxifen treatment of spontaneous carcinomas/adenomas in vivo (Fig. 4G). We chronically treated MMTV-PyMT tumours by implantation of MMTV-PyMT Rac1-FRET mice with tamoxifen pellets (5 mg/pellet, 60-day release), at the point at which the tumours became palpable. Tamoxifen treatment significantly slowed tumour growth in vivo, as previously shown[45] (Fig. 4H). Upon the development of tamoxifen treated MMTV-PyMT primary tumours of around 120mm³ in size in the inguinal mammary fat pad, mice were implanted with mammary optical windows, allowing us to assess Rac1 activity in real time in the context of the tumour microenvironment. We observed that Rac1 activity was not uniform across the tissue, but instead there was heterogeneity, with regions of high and low activity within tumours chronically treated with tamoxifen (Fig. 4I). These in vivo data were consistent with the mixed population that we detect following development of resistance in MCF-7 cells, where a proportion of cells have high expression of a RAC1 node following the development of resistance to endocrine therapy (Fig. 3).

## P-Rex1 is high in ER+ metastases, and associated with late recurrence

Since Rac1 signalling was elevated in ER+ Endocrine Tolerant cells and tamoxifen treated MMTV-PyMT cells, we examined the association between the Rac1 pathway proteins and luminal type breast cancers. P-Rex1 protein was significantly higher in luminal type breast cancers than normal tissue (Fig. 5A), but Rac1 and Pak2 proteins did not vary significantly (Supplementary Fig. S6A/B). This was distinct to the profile of proliferative marker Ki67 and the marker we identified in Fast growing cells, EGFR, which were both significantly lower in luminal type cancers than normal tissue or triple negative breast cancers (TNBC) (Fig. 5B/C).

We then examined expression of *PREX1*, *RAC1*, *PAK2* and *EGFR* in scRNAseq from 26 breast cancers including 11 ER+ breast cancers to determine the co-occurrence of gene expression within epithelial cancer cells. *PREX1* co-occurred with *ESR1* expressing cells of the Luminal A and Luminal B subtypes, whereas *EGFR* was mainly expressed in Her2 and TNBC type cancer cells (Fig. 5D). Cycling cells had high levels of *MKI67* and *EGFR*, but were notably lacking *PREX1* or *ESR1*

expression. We then performed scRNAseq on an ER+ lung metastasis from a patient that presented with ER+ /PR+ disease and had progressed on tamoxifen, aromatase inhibitors and chemotherapy (Fig. 5E). While only a small subset of epithelial cancer cells was identified in this sample by *EPCAM/KRT18* expression, a cluster were positive for *ESR1* expression (Fig. 5E). *ESR1* positive cells showed high expression of *PREX1*, *PAK2* and *RAC1*, but were predominantly negative for *EGFR* and *MKI67* (Fig. 5E, Supplementary Fig. S6C). There was also a significant positive correlation between *ESR1* expression and *PREX1* expression within individual epithelial cells of the metastasis when both genes are detected (Fig. 5F). Matched normal lung epithelium lacked *ESR1* and *PREX1* expression but showed *RAC1* and *PAK2* expression (Fig. 5E, Supplementary Fig. S6C).

We next examined the association of *PREX1*, *RAC1*, *PAK2*, *EGFR* and *MKI67* with survival across endocrine therapy treated breast cancer. We used composite normalised datasets with follow-up of 16.25 years, which we split into 0-5 years (early) and 5-16.25 years (late) to compare the risk of early and late recurrences. Within the first 5 years of endocrine therapy, there was no difference in recurrence-free survival of patients with high/low *PREX1* expression, however recurrence-free survival was lower in patients with high *PREX1* after 5 years (p < 0.075; Fig. 5G). High expression of *RAC1* or *PAK2* showed an association with poor recurrence-free survival throughout recurrence (Supplementary Fig. S6D). We compared the recurrence-free survival of patients who are likely to have highly proliferative cancers, and *MKI67* was predictive of poor-recurrence free survival in the first five years following primary diagnosis, but it was not at all predictive of late recurrences (Fig. 5G). *EGFR* was not predictive for early or late recurrence (Supplementary Fig. S6D).

Since there was a trend for *PREX1* mRNA to be associated with late recurrence following endocrine therapy (Fig. 5G) but *PREX1* was expressed early following fulvestrant treatment (Fig. 3K/L), we determined in a retrospective cohort with late recurrences and up to 25 years of follow-up whether P-Rex1 protein could act as a prognostic marker of late recurrence. P-Rex1 protein was examined by IHC in this cohort of ER+ cancers that progressed to gynaecological metastases with a median time of 5 years[46] (Fig. 5H, Supplementary Fig. S6E).

High P-Rex1 was significantly associated with late detection (median of 7 years) of metastatic recurrences compared to low P-Rex1 (median of 4.5 years) (Fig. 5I). However, there was no relationship between P-Rex1 expression and time to breast cancer specific death in the cohort of patients (Fig. 5J). This suggests that while recurrences from primary tumours with high P-Rex1 occur later, these patients do not have prolonged survival.

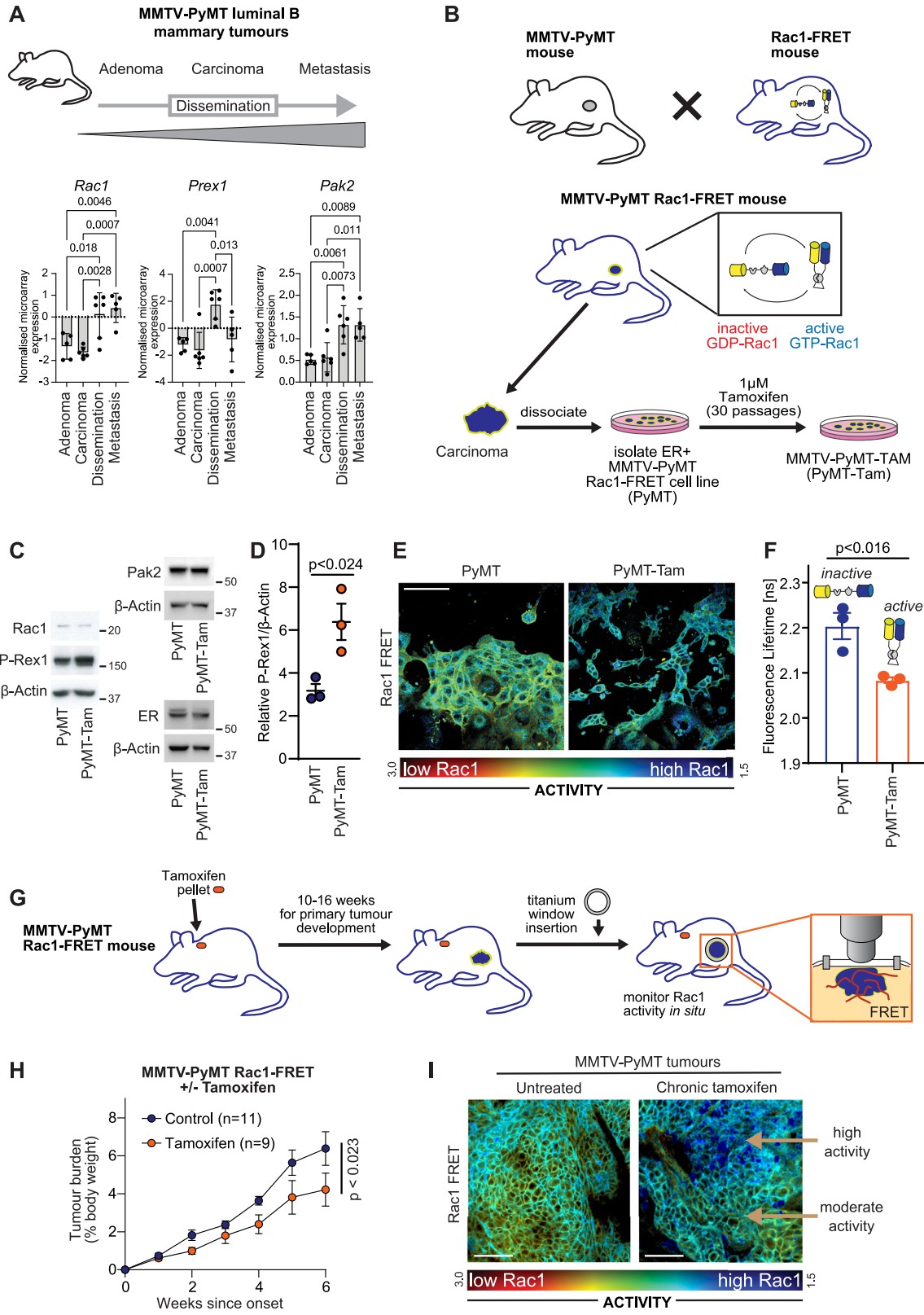

We then examined how P-Rex1 expression related to ER expression by examining whether metastases that maintain ER expression also maintain P-Rex1 expression. We found that ER+ metastases generally maintained or gained P-Rex1 expression (Fig. 5K). Six matched metastases from primary ER+ tumours had reduced ER expression, and these metastases generally showed significantly lower P-Rex1 expression in the metastases than the primary tumour (Fig. 5L). Overall, these

data suggest that ER+ metastases maintain P-Rex1 expression and active Rac1 signalling, which could be a driver of disease spread.

## Targeting of Rac1 activity in the context of endocrine therapy resistance

Since Rac1 signalling is elevated in slow proliferating and late recurring ER+ disease, we next assessed whether Rac1 inhibition is a viable

**Fig. 4 | Rac1 activity in a MMTV-PyMT model of tamoxifen resistance. A** Schematic of spontaneous tumour development and dissemination in the MMTV-PyMT model. Expression of *Rac1* (A_51_P513254), *Prex1* (A_51_P348372) and *Pak2* (A_51_P172323) in stages of MMTV-PyMT development (GSE43566[43]) of adenoma (n = 5), carcinoma (n = 6), dissemination (n = 6) and metastases (n = 5). Data analysed by one-way ANOVA with Tukey's multiple comparisons. **B** Schematic of derivation of MMTV-PyMT Rac1-FRET biosensor mice with biosensor in active (blue) and inactive (red) conformations, and derivation of a chronic tamoxifen biosensor cell line. **C** MMTV-PyMT Rac1-FRET parental (PyMT) and chronic tamoxifen (PyMT-Tam) cell lines western blotted for ER, P-Rex1, Pak2 and Rac1. β-actin loading control is shown for each western blot. Representative of three replicates. **D** Densitometry of P-Rex1 expression normalised to β-actin from triplicate biological replicates, analysed by two-sided t-test, and error bars represent SEM. **E** Representative image of Rac1-FRET activity in MMTV-PyMT cells chronically treated with tamoxifen with matched parental control. Scale bar = 100 μm. **F** Quantitation of Rac1-FRET activity of cells in E. Triplicate biological replicates analysed by two-sided t-test; error bars represent SEM. **G** Schematic of MMTV-PyMT Rac1-FRET biosensor mouse with titanium optical imaging window implanted over a mammary tumour. **H** Tumour growth of control (n = 11) and tamoxifen (n = 9) treated MMTV-PyMT Rac1-FRET mice. Data analysed by two-way ANOVA with a mixed effect model. Error bars represent SEM. **I** Image of Rac1-FRET activity in control and tamoxifen treated mammary tumours, representative of three biological replicates. Scale bar = 50 μm. Source data are provided as a Source Data file.

option in this setting using two proof-of-principle Rac1 inhibitors, NSC23766 and 1A-116, and R-ketorolac (Fig. 6A). NSC23766 is a small molecule inhibitor that blocks interaction of Trio and Tiam with Rac1[47] and 1A-116 blocks the interaction between P-Rex1 and Rac1[48]. R-ketorolac is the R-enantiomer in the racemic mix of the analgesic ketorolac, where S-ketorolac is used therapeutically as an NSAID (nonsteroidal anti-inflammatory drug), but R-ketorolac has off-target inhibition of Rac1[49]. Of these drugs we found that Endocrine Tolerant cells were more sensitive to growth inhibition by NCS23766 than Fast-growing cells, but there were no differences in growth inhibition by 1A-116 or R-ketorolac in each of the cell lines (Supplementary Fig. S7A-C).

Given that Rac1 activity is particularly implicated in cell migration and survival, we assessed the effects of these inhibitors on Fast-growing and Endocrine Tolerant cell lines in survival and scratch wound assays. For survival assays, the cells were plated at low density and treated with vehicle, NSC23766, 1A-116 and R-ketorolac for 1 week (Fig. 6B/C). Overall, the Endocrine Tolerant cells formed smaller colonies than Fast-growing cells, consistent with their growth kinetics. However, the Endocrine Tolerant cells showed significantly reduced colony formation with both R-ketorolac and NSC23766, whereas there was only a trend towards reduced colony formation in the Fast-growing cells with R-ketorolac treatment. Endocrine Tolerant and Fast-growing cells were next grown to confluence and scratch wound closure measured for 24 hours (Fig. 6D/E, Supplementary Fig. S7D/E). Surprisingly, even though Endocrine Tolerant cells have enhanced Rac1 pathways, we found that the cells migrated considerably slower in these assays. However, wound closure of Endocrine Tolerant cells was significantly inhibited by NSC23766 and 1A-116, with a trend to inhibition with R-ketorolac. Fast-growing cells did not show an inhibition of wound closure with any drug.

We next assessed the efficacy of these drugs in vivo using tamoxifen treated mammary tumours of MMTV-PyMT:Rac1-FRET mice (Fig. 6F). Mice were treated with tamoxifen to inhibit tumour development and windows were implanted over these tamoxifen resistant tumours (~120 mm³ tumours). Animals were then treated with a single dose of 4 mg/kg NSC23766 or 1 mg/kg R-ketorolac and imaged for up to 96 hours, during which time Rac1 activity was monitored by FRET. Animals treated with NSC23766 showed acute changes to Rac1 signalling in tumours within 2 hours of drug administration (Fig. 6G), which was sustained for at least 6 hours. Treatment with R-ketorolac demonstrated reduced Rac1 signalling between 24-48 hours after administration (Fig. 6H).

Finally, we assessed long-term response of a patient derived xenograft (PDX) model to NSC23766 or R-ketorolac as single agents, or when combined with tamoxifen. We first screened eight breast cancer PDX models for P-Rex1 using IHC (Supplementary Fig. S8A, Supplementary Table 1). Of these models, the KCC4653 PDX model had the highest expression of P-Rex1. The KCC4653 model was derived from a liver metastasis of an ER+ breast cancer patient who had progressed on three types of endocrine therapies (tamoxifen, letrozole, exemestane), and a range of chemotherapies (Fig. 7A).

Following tumour implantation and expansion, the KCC4653 model was treated with Vehicle, NSC23766, R-ketorolac, tamoxifen, NSC23766 + tamoxifen or R-ketorolac + tamoxifen (Fig. 7B). Treatment with single arm R-ketorolac led to a significant decrease in tumour volume (Fig. 7C), and treatment with either NSC23766 or R-ketorolac led to a significant increase in survival (6 weeks for NSC23766, 6.5 weeks for R-ketorolac, versus 5.25 weeks for vehicle treatment, Fig. 7D). Tamoxifen treatment led to a significant retardation of tumour growth as a single agent, however, tumours continued to slowly increase in size, consistent with the patient history of progression on tamoxifen (Fig. 7C). By contrast, the combination of tamoxifen with NSC23766, or tamoxifen with R-ketorolac led to tumour regression (Fig. 7C). The cohort was terminated at 18 weeks at the end of treatment, at which time there was a significant decrease in tumour weight for those mice treated with either tamoxifen with NSC23766, or tamoxifen with R-ketorolac, when compared to tamoxifen alone (Fig. 7E). Toxicity assessments of treatment via blood counts showed that while there were elevated counts of neutrophils and monocytes associated with tamoxifen treatment, there were no short or long-term effects noted with NSC23766 or R-ketorolac alone (Supplementary Fig. S9A-C). The combination of NSC23766 or R-ketorolac with tamoxifen did not lead to significant changes in red blood cell counts (Supplementary Fig. S9D), however there was a slightly elevated number of platelets following prolonged treatment with tamoxifen and R-ketorolac (Supplementary Fig. S9E).

In earlier experiments we had seen that Endocrine Tolerant cells, which have high expression of P-Rex1, had altered collagen deposition when grown as xenografts. For this reason, we examined collagen in tumours that had been treated with Rac1 inhibitors +/- tamoxifen. Using picrosirius red staining, we saw that the tumours of mice treated with Rac1 inhibitors did not show a significant change compared to vehicle, although some individual tumours had higher staining (Fig. 7F/ G). However, all long term treatment arms showed a significant increase in complex collagen structures compared to vehicle. Notably, it was only the combination therapies, and not tamoxifen alone, that showed a greater accumulation of collagen than single agent Rac1 inhibitors (Fig. 7G). Thus, the inhibition of Rac1 activity does not prevent the deposition of collagen, although it may be altering the remodelling of collagen fibres.

While R-ketorolac has only been prospectively assessed in an exploratory trial of ovarian cancer[50], racemic ketorolac and other analgesics are routinely used peri-operatively for pain management of breast cancer surgeries. An association between NSAID use, particularly ketorolac, and reduced cancer recurrence has been documented[51]. We separated studies of peri-operative analgesics into those with known GTPase activity (ketorolac) and without GTPase activity (diclofenac, sufentanil[49]) and examined the association of ketorolac intervention with breast cancer recurrence (Supplementary Table 2). These cohorts included 71-85% ER+ cancers and 26-84 months median follow-up. Peri-operative ketorolac was associated with a significantly reduced risk of breast cancer recurrence (HR:0.50 CI 0.32-0.80; P < 0.004 (Fig. 8A)) whereas non-GTPase analgesics, including

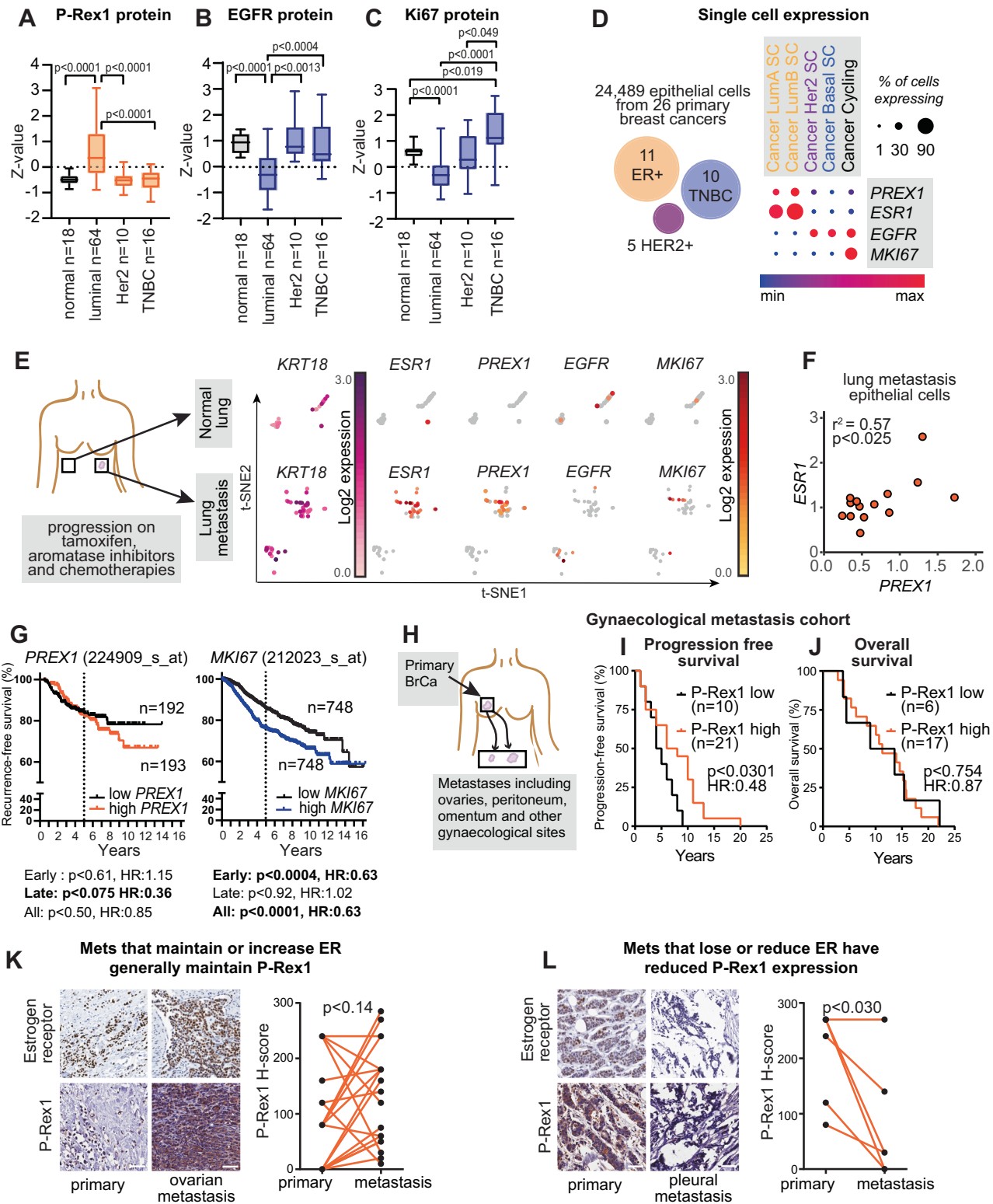

**K** Mets that maintain or increase ER generally maintain P-Rex1

**L** Mets that lose or reduce ER have reduced P-Rex1 expression

the NSAID diclofenac, did not associate with reduced recurrence (Fig. 8B, Supplementary Table 2). Overall, it appears that peri-operative ketorolac usage is associated with reduced breast cancer recurrence.

## Discussion

Late recurring breast cancer is a significant cause of breast cancer related death. Current signatures for late recurrence often lack proliferative markers[52], and some proliferative molecular signatures are only predictive for early recurrences of ER+ breast cancer[53]. In keeping

with those observations, we found that a cohort with average follow-up of 24 months had metastases with lower average proliferation than the matched primary ER+ tumours[9]. While the fast proliferation of metastasis is sometimes conflated with lethality of disease, slow proliferating cancers have similar associated mortality to rapidly growing cancers based on survival analyses of patients detected by interval screening[54]. Thus, drug resistance models with a low proliferative rate could be a crucial resource for the identification of novel predictive markers and therapeutic targets in late recurrence.

**Fig. 5 | Rac1 and P-Rex1 pathways in late recurring breast cancers.** Protein expression from CPTAC[122] across normal breast (*n* = 18), luminal type breast cancers (*n* = 64), Her2 (*n* = 10), triple negative (TNBC; *n* = 16) for **A** P-Rex1, **B** EGFR and **C** Ki67. Data analysed by t-tests as described in[122]. **D** Expression matrix for *PREX1*, *ESR1*, *EGFR* and *MKI67* in epithelial cells from 26 primary breast cancers (*n* = 24489 cells), in cells subclassified as Luminal A (*n* = 7742), Luminal B (*n* = 3368), HER2 (*n* = 3708), Basal (*n* = 4312) or Cycling (*n* = 5359). Heat scale indicates range of expression from 0-100%, and symbol size indicates the percentage of cells that express the transcript. **E** scRNAseq analysis of epithelial cells from a lung metastasis (*n* = 51) and normal lung (*n* = 27) from a breast cancer patient previously treated with tamoxifen, ovarian suppression, aromatase inhibitors and chemotherapies. Epithelial cells were identified by *KRT18* expression (pink). tSNE plots shown for *ESR1*, *PREX1*, *EGFR* and *MKI67* within epithelial cell subsets. **F** Correlation between *ESR1* and *PREX1* expression in *ESR1/PREX1* positive cells of lung metastasis (*n* = 15 cells) shown in (E), analysed by Pearson correlation. **G** Recurrence-free survival of patients with high (*n* = 193) and low (*n* = 192) *PREX1* expression (224909_s_at), or high (*n* = 748) and low (*n* = 748) *MKI67* expression (212023_s_at), treated with tamoxifen and aromatase inhibitors, divided into early (0-5 yrs) and late (5-16.5 yrs) recurrence. Data analysed by Log-rank (Mantel-Cox) test with hazard ratio. **H** Schematic of primary ER+ breast cancer (BrCa) cohort metastasising to gynaecological organs. **I/J** Time from primary tumour to **I** metastasis or **J** death based on P-Rex1 expression in a breast cancer cohort with gynaecological metastases. Patients were stratified based on low (H-score ≤ 50) and high (H-score > 50) P-Rex1 expression. Data analysed by Log-rank (Mantel-Cox) test with hazard ratio. **K** IHC analysis of ER and P-Rex1 expression in matched primary and metastatic (*n* = 19) samples that maintain or gain ER expression in the metastatic lesion. Data analysed by paired two-sided t-test. Scale bar = 100 μm. **L** IHC analysis of ER and P-Rex1 expression in matched primary and metastatic (*n* = 6) samples that lose or reduce ER expression in the metastatic lesion. Data analysed by paired two-sided t-test. Scale bar = 100 μm. Source data are provided as a Source Data file.

We identified upregulation of endogenous Rac1 activity in slow-growing models of chronic hormone therapy treatment. Elevated Rac1 activity has not been previously documented in the context of hormone therapy resistance, although experimental models with ectopic expression of Rac1 or Rac1 pathway activators (Tiam1, Vav, IGFR) have demonstrated that elevated Rac1 can promote hormone therapy resistance[31,55–57]. We identified a key upregulated target in hormone therapy resistance, P-Rex1, which is an ER target gene[58], and is important for cell migration in Endocrine Tolerant cells (Fig. 3P/Q). P-Rex1 is strongly associated with *ESR1* expression and luminal breast cancer as shown by our data, and the data of others[21,37,59]. We also identified that P-Rex1 expression is maintained in ER+ gynaecological metastases in a cohort of ER+ patients. While P-Rex1 is expressed in primary cancers that go on to metastasise[21,60] and in an ER+ model of metastatic bone dissemination[21], there are conflicting reports on the association of P-Rex1 with outcome in ER+ breast cancer. High P-Rex1 protein[61], *PREX1* mRNA[62] and lack of *PREX1* methylation[59] are associated with poor disease-free survival across multiple cohorts, although not in the Metabric cohort[58]. In a further cohort of 121 primary breast cancers, high P-Rex1 protein was associated with improved disease-free survival, but this cohort was enriched for ER- cases (46.3%) and the follow-up was short at only 29 months[63]. Overall, most survival studies are consistent with our findings in a cohort of patients with ER+ breast cancer with 25 years follow-up that high P-Rex1 is associated with late recurrence of disease.

The upregulation of P-Rex1 and Rac1 signalling in primary and metastatic ER+ disease provides new opportunities for therapeutic targeting. We used three inhibitors to investigate this potential. Two inhibitors, 1A116 and NSC23766, are proof of principle Rac1 inhibitors, and ketorolac is a clinically used NSAID with off target Rac1 inhibitor action. With in vitro cell line assays, we showed that the three Rac1 inhibitors were able to reduce colony formation and cell migration. Real-time biosensor analysis in vivo demonstrated a reduction of Rac1 activity in chronically tamoxifen treated tumours following NSC23766 and R-ketorolac treatment. Furthermore, treatment of an ER + PDX model derived from a patient who has failed endocrine therapy showed that the combination of NSC23766 or R-ketorolac with tamoxifen was able to significantly reduce tumour burden.

Of the Rac1 inhibitors, R-ketorolac is administered clinically in a racemic mix with S-ketorolac, which is a Cox2 inhibitor and NSAID. By meta-analysis we show that patients that received ketorolac during breast cancer surgery have reduced breast cancer recurrence, whereas those that receive analgesics with no known GTPase activity do not. This could be due to the anti-inflammatory actions of S-ketorolac or a combined effect of GTPase inhibition and anti-inflammatory activity leading to a reduction in metastatic seeding[50]. Caveats of this analysis are that patients in these cohorts were not solely ER+, some received a variety of analgesics, and there are a limited number of studies, which

are all factors that could affect or bias the analysis. Long-term prospective clinical trials are needed to definitively show a benefit of ketorolac to reduce recurrence. However, we additionally show that the pure R-enantiomer of ketorolac is able to reduce tumour growth of an endocrine therapy resistant PDX model, as well as showing enhanced anti-tumour effects when combined with tamoxifen. This is consistent with its ability to reduce primary tumour development in MMTV-PyMT mice[64] and reduce omental engraftment and tumour burden of xenografted ovarian cancer cells[65]. Overall, R-ketorolac has exciting clinical potential to be applied therapeutically to reduce ER+ metastatic spread and burden.

Our data showed that a high Rac1 signalling signature and high P-Rex1 are generally mutually exclusive with EGFR signalling. A computational modelling approach has also identified a lack of Rac1 signalling downstream of EGFR/PI3K signalling in hepatocytes[66]. While P-Rex1 can activate PI3K signalling[37], our data suggests that this does not hinge on EGFR. Interestingly, reliance upon EGFR signalling has been frequently identified across in vitro and pre-clinical models of hormone therapy resistance[67–72] but this has failed to validate in clinical settings: EGFR does not predict reduced recurrence-free survival of tamoxifen treated patients in large randomised trials with 10-20 years follow-up[73–75], and EGFR inhibitors have not led to consistent improvements in trials of advanced hormone therapy refractory ER+ disease[76–78]. One explanation for the failure of translation is that pre-clinical modelling typically selects for faster growing clonal populations with enhanced growth factor receptor signalling rather than maintaining the representative resistant population. Alternatively, this could reflect a paradoxical loss of EGFR function in advanced breast cancers[79].

Overall, our approach has highlighted the potential for slow growing therapy tolerant cells to persist in patients and play a major role in late breast cancer recurrence. A strong parallel can be drawn from the study of "drug tolerant persister" cells (DTPs) in other cancer types, which are drug refractory cells often characterised by slow cycling. A distinction in our study is that we do not observe obvious enrichment for stem cell markers in our drug tolerant cells, compared to the heightened CD44, ALDH1, CD133 and CD271 that are often observed in other DTPs[80]. We also do not see strong evidence that the cells arise from a precursor population in the parental cells, although a barcoding approach would be needed to show this definitively. Instead the drug tolerant cells that we identify are enriched for signatures of cells that enter a senescent-like state post-treatment, although this too can be a feature in cancer stem cells[80]. In vivo the Endocrine Tolerant cells maintained their large nuclear morphology and were associated with increased deposition of collagen, which are frequently observed features of senescent cells[81].

The relationship between Endocrine Tolerant cells and previously characterised dormant cells also needs to be considered. One study

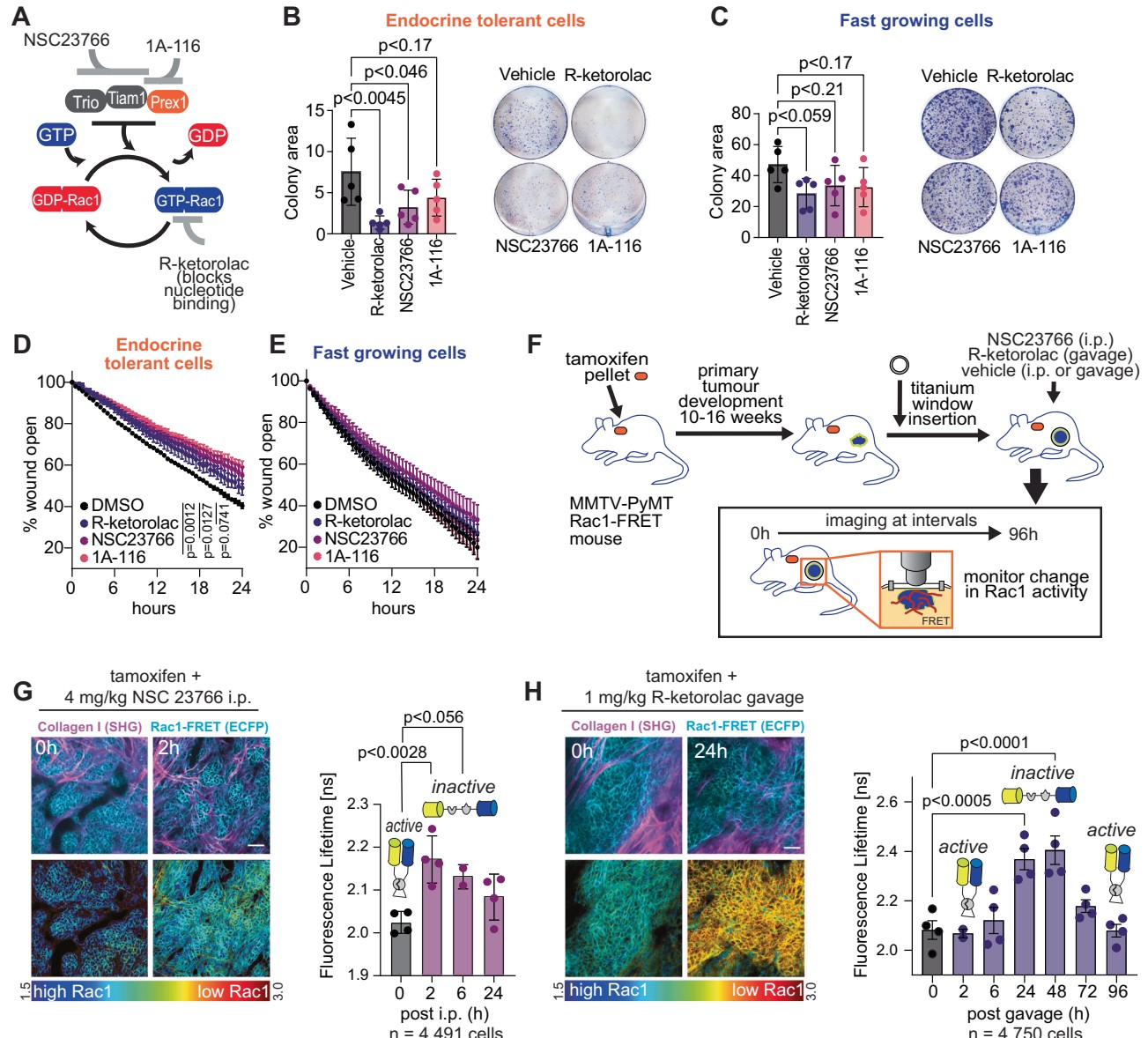

**Fig. 6 | Targeting Rac1 pathway in endocrine therapy resistant cancers. A** The Rac1 pathway can be targeted by small molecule inhibitors; NSC23766, 1A-116, R-ketorolac. **B/C** Endocrine Tolerant and Fast-growing cells were treated with DMSO, 100 μM R-ketorolac, 25 μM NSC23766 and 6.25 μM 1A-116 for 1 week, and colony formation detected with 10% Diff Quik Stain 2. Colony formation (% colony area) ($n = 5$ biological replicates) was quantitated using Image J. Data analysed by one-way ANOVA with multiple comparisons. Error bars are SEM. **D/E** Wound closure in Endocrine Tolerant and Fast-growing cells treated with DMSO, 100 μM R-ketorolac, 25 μM NSC23766 and 6.25 μM 1A-116 for 24 hours. Data ($n = 3$ biological replicates) analysed by two-way repeated measure ANOVA for each treatment versus vehicle. Error bars are SEM, or smaller than the symbol. **F** Schematic of tamoxifen resistant MMTV-PyMT:Rac1-FRET biosensor mouse with titanium window implanted over mammary tumour to monitor Rac1-FRET activity after

treatment with NSC23766 (4 mg/kg; intraperitoneal injection) or R-ketorolac (1 mg/kg; oral gavage). **G** Representative image of Rac1-FRET activity in tamoxifen resistant (TamR) MMTV-PyMT tumours treated with a single injection 4 mg/kg NSC23766, and single cell based temporal quantitation of Rac1-FRET activity of tumour. Data ($n = 4$ mice measured at 0, 2, 24 h; $n = 2$ at 6 h) analysed by one-way ANOVA with Dunnett's multiple comparison test, error bars are SEM. **H** Representative image of Rac1-FRET activity in tamoxifen resistant (TamR) MMTV-PyMT tumours treated with 1 mg/kg R-ketorolac and imaged for up to 96 hours, and single cell based temporal quantitation of Rac1-FRET activity of tumour. Data ($n = 4$ mice measured at 0, 6, 24, 48, 72, 96 h; $n = 2$ at 2 h) analysed by one-way ANOVA with Dunnett's multiple comparison test, error bars are SEM. Source data are provided as a Source Data file.

found that dormant breast cancer cells lack senescence features[82], which is consistent with our observations of a senescent-like, but not dormant, phenotype in slowly proliferating Endocrine Tolerant cells. We also compared the gene signature of Endocrine Tolerant cells with the gene expression profiles associated with an ER+ dormancy model[26] (Supplementary Fig. S3), but our analysis did not reveal significant similarities. However, it is noteworthy that *PREX1* was identified as one of the 25 genes common to Endocrine Tolerant cells and the dormancy

model. This overlap suggests that the Rac1 pathway may play a significant role in cells that have entered a more profound state of cell cycle arrest, such as that associated with dormancy.

A unique feature of Endocrine Tolerant cells compared to previously characterised breast cancer stem cells is that they maintain estrogen receptor and estrogen responsiveness[83]. Other studies on recurrent ER+ metastases have also identified *ESR1* expression within disseminated cells[84]. This is consistent with

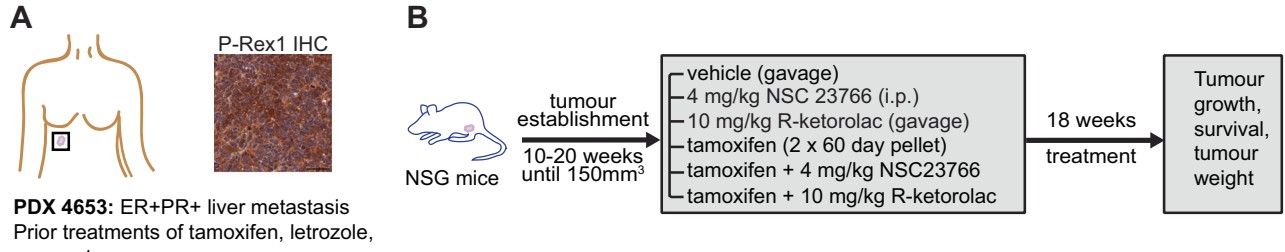

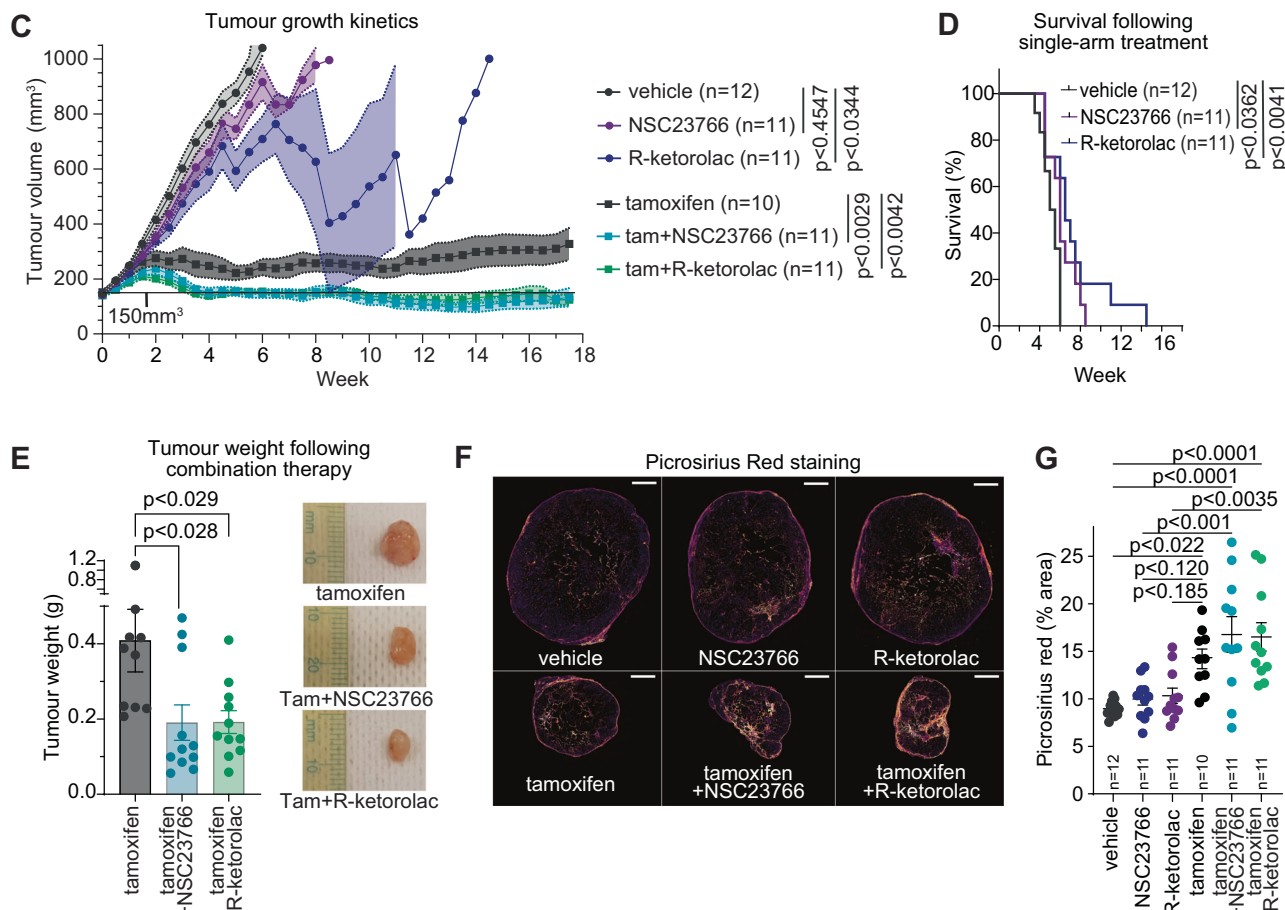

**Fig. 7 | Targeting Rac1 pathway in endocrine therapy resistant cancers. A** Schematic of PDX model KCC4653, and IHC staining of tumour tissue for P-Rex1. Scale bar is 50 μm. IHC image is identical to IHC image in Supplementary Fig. S8A, as these images were used to screen for tumours high for P-Rex1. **B** Schematic of treatment of NSG mice. Following tumour implantation and expansion to ~150mm³, 10-12 mice/arm were treated with (i) Vehicle (gavage), (ii) 4 mg/kg NSC23766 (intraperitoneal injection, i.p.), (iii) 10 mg/kg R-ketorolac (gavage), (iv) Tamoxifen (60 day slow release 5 mg pellet), (v) Tamoxifen + 4 mg/kg NSC23766 and (vi) Tamoxifen + 10 mg/kg R-ketorolac. Tamoxifen pellets were reimplanted halfway through treatment. **C** Tumour growth was measured by callipers. Outcomes were short-term at ethical endpoint (vehicle, NSC23766, R-ketorolac) and long-term at end of treatment (tamoxifen, tamoxifen + NSC23766, tamoxifen + R-ketorolac). Statistical analyses were performed for short and long-term endpoints separately, using mixed effect analysis and Tukey's multiple comparison testing. Error bars represent SEM. Individual tumour growth trajectories are shown in Supplementary Fig. S8B. **D** Survival analysis of tumour-bearing mice treated with vehicle, NSC23766, R-ketorolac, performed using Logrank test. **E** Final tumour weight of mice treated with tamoxifen ($n = 10$), tamoxifen + NSC23766 ($n = 11$) or tamoxifen + R-ketorolac ($n = 11$), following 18 weeks of treatment. Analysed by one-way ANOVA with Tukey's multiple comparison test, error bars represent SEM. Representative examples of tumours are shown. **F** Picrosirius red staining of endpoint tumours. Scale bar is 20 mm. **G** Quantitation of Picrosirius red staining as % area of tumour ($n = 10$-12 samples per arm, as indicated on the graph). Error bars represent SEM and statistical analysis performed by one-way ANOVA with Tukey's multiple comparison test. Source data are provided as a Source Data file.

~80% of ER+ recurrences sustaining ER expression[85], and the higher frequency of ER retention in late recurrences[86]. The identification of these slow growing cells in resistance increases our understanding of metastatic ER+ breast cancer and identifies driver pathways such as Rac1 signalling, which ultimately creates the potential for different therapeutic interventions.

## Methods

### Ethics approval

All research procedures involving animals were conducted in accordance with ethical guidelines. Animal studies were performed with ethics approval (ARA 16/13, 17/23, 19/10, 19/13, 21/09, 24/16) from the Garvan/St Vincents Animal Ethics Committee, Sydney, Australia.

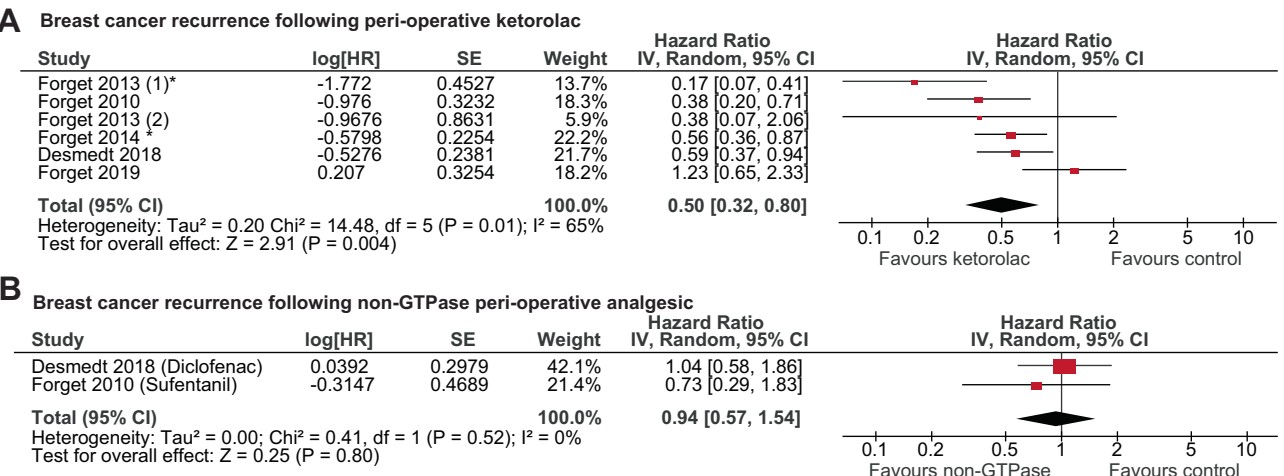

**Fig. 8 | Analysis of recurrence free survival of breast cancers following ketorolac usage.** Meta-analysis of studies of recurrence-free survival associated with peri-operative analgesic use for breast cancer. Studies separated into studies with **A** Ketorolac, and **B** other analgesics including sufentanil and the NSAID diclofenac. Ketorolac treatment studies indicated with (\*) had ketorolac treatment for the majority of patients, but included data from a subset of patients treated with diclofenac. Data was analysed using a random effects model, and was performed using Revman 5.0 software[126]. HR is Hazard Ratio, SE is standard error, CI is confidence interval. Bars represent the confidence interval, and the central symbol indicates the hazard ratio, where the centre of the symbol is the value of the hazard ratio in the adjoining table. Size of red rectangle indicates the relative weight of the study, as indicated in the adjoining table. Studies are listed in Supplementary Table 2.

Human samples were collected through the MonARC Programme, approved human ethics protocol SVH17/173 at St Vincent's Hospital, Sydney Australia, and analysed under the Personalised Medicine for Breast Cancer study, protocol x19-0496.

### Cell culture and derivation of cell lines
MCF-7 cells were cultured in phenol red free RPMI1640 supplemented with 5% charcoal stripped FBS (csFBS, method adapted from[87]), 1% penicillin/streptomycin, 10 pM 17-β estradiol (Sigma Aldrich; resuspended in EtOH) with a media change every 3-4 days.

To derive cell lines with different proliferative capacity, MCF-7 cells treated with 100 nM fulvestrant (Sigma Aldrich; resuspended in EtOH) for 20 days had their media removed, and were washed once with prewarmed phosphate buffered saline (PBS) to remove all traces of serum. 2 μM CellTrace (Thermo Fisher) was added and cells incubated in a $CO_2$ incubator for 20 minutes. The CellTrace solution was removed, and cells washed twice with medium with 1% csFCS, followed by addition of culture media plus 100 nM fulvestrant. Cells were cultured for a further 12 days until cell sorting. Cells were then trypsinised, and resuspended in sorting buffer (PBS + 2% FCS + 1.5 mM EDTA + 2 x Penicillin/Streptomycin + 2 x Antibody/Antimycotic) prior to sorting into low, medium and high fractions based on relative CellTrace incorporation.

After 3 months in culture, the highest CellTrace expressing population from the first sort was re-labelled with 2 μM CellTrace dye, and FACS sorted based on the CellTrace expression levels. After another 6 months in culture this process was repeated on the highest CellTrace expressing population from the second sort. Cells were cultured for a further 12 months until they reached a constant doubling time. All subsequent assays were performed in the presence of 100 nM fulvestrant, except as indicated.

### Generation of *PREX1* knockdown cells
MCF-7 cell line derivatives (low and high CellTrace fractions) were transduced with Mission Lentiviral transduction particles (Sigma-Aldrich) containing either a pLKO.1-Puro plasmid encoding non-targeting shRNA, or pLKO.1-Puro plasmid encoding shRNA targeting *PREX1* mRNA (construct TRCN0000044796). Stable cell lines were established by puromycin selection (0.25-1 g/L; Sigma-Aldrich).

### Hormone response assays
Cells were treated with the following drugs and reagents: 17-β estradiol (Sigma Aldrich; resuspended in EtOH), fulvestrant (Sigma Aldrich; resuspended in EtOH), tamoxifen (Sigma-Aldrich; resuspended in THF), estrogen deprivation (phenol red free RPMI1640 + heat-inactivated csFBS). Live-cell imaging of cell number was performed using an Essen Bioscience IncuCyte ZOOM Live-Cell Analysis System. Cells were imaged over 5 days every 1–2 days over 4 fields of view per well. Images were analysed using IncuCyte ZOOM Software Version 2016B.

### ß-galactosidase assays
Senescence was assayed by visualising ß-galactosidase activity along with cell morphology using Senescence ß-Galactosidase Staining Kit as per the manufacturer's instructions (Cell Signalling Technology). 3-6 fields of view were imaged per replicate, and the proportion of senescent cells scored with the ImageJ Cell Counter. Cells were scored as senescent when they exhibited blue ß-galactosidase stain, as well as cellular morphology consistent with senescence including perinuclear ß-galactosidase.

### Growth factor stimulation assays
Cells seeded at $2 \times 10^5$ cells per well in a 6 well plate were stimulated with 100 ng/mL EGF or 100 nM 17-ß estradiol for 30 mins. Alternatively, cells were serum starved (0.1% csFBS) for 24 hours and stimulated with 10% FBS for 30 mins. Lysates were then analysed by MAGPIX assays.

In addition, cells were serum starved with 0.1% csFBS for 24 hours prior to stimulation with either DMSO, 100 ng/mL EGF (R&D Systems), 1 − 100 nM insulin (Novo Nordisk) or 10 ng/mL IGF1 (Thermo Fisher Scientific) for up to 30 minutes before cells were harvested on ice. Cells were lysed with RIPA buffer for 10 min on ice prior to brief vortexing and then centrifuged at 10,000 rpm to clear the lysates. Supernatants were collected and protein concentration determined by Bradford Assay.

## Immunoblotting and Magpix protein detection

Total cell lysates were separated using 4–12% polyacrylamide gels (Invitrogen, or homemade), prior to transfer to PVDF membranes. Primary antibodies were cyclin A (1:1000, #sc-239), cyclin D1 (1:500, #sc-20044), estrogen receptor-α (human) (1:500, #sc-543), β-actin (1:15,000, #sc-69879), and GAPDH (1:15,000, #sc-32233) from Santa Cruz Biotechnology; p21 (1:1000, #610234), and total Rb (1:500, #554136) from BD Biosciences; P-Rex1 (1:1000, #HPA001927) from Sigma-Aldrich; Rac1 (1:1000, #ARC03) from Cytoskeleton Inc; PAK2 (1:2000, #A4553) from AB Clonal; estrogen receptor-α (mouse) (1:1000, #ab32063) from Abcam; EGFR (1:1000, #H00001956-M02) from Abnova; pHER2 (Tyr1221/1222) (1:1000, #2243); pEGFR (Tyr1045) (1:1000, #2237); HER2 (1:1000, #2165) and pERK (Thr202/Tyr204) (1:1000, #9101) from Cell Signalling Technology; and pIR/IGF1R (Tyr1162/Tyr1163) (1:1000, #44-804 G) from Invitrogen.

Membranes were cut into smaller molecular weight ranges to allow matched blots of proteins of different molecular weight. Chemiluminescence was performed with the Fusion FX7 Imaging System or Konica Minolta SRX101A processor. Densitometry of western blots was performed using ImageJ/FIJI[88].

Multiplex analysis of phospho-proteins was performed using the Bio-Plex MAGPIX system (#171015044) and Bio-Plex Pro-Wash Station (Biorad). Cell lysates were prepared as in[89] and analysed according to manufacturer's instructions using the Milliplex Map RTK Phosphoprotein Magnetic Bead panel – Cell Signalling Multiplex Assay – 9 Plex (HPRTKMAG01K). Data was generated with Bio-Plex Manager MP and analysed on the Bio-Plex Manager 6.1 software. Relative phosphoprotein was determined from triplicates.

## Scratch wound assays

Cells were seeded at $6\times10^5$ cells per chamber in 2-well culture slides with insert (80209, Ibidi). After 24 hours, the inserts were removed to create a uniform wound within the confluent cell monolayer then treated with 100 μM R-ketorolac (MedChemExpress), 25 μM NSC23766 (Selleck Chemicals), 6.25 μM 1A-116 (MedChemExpress) or DMSO. Drug doses were selected to be - IC25 dose, or 100 μM for R-ketorolac which did not show any proliferative inhibition in a dose response assay. Using phase contrast on the Leica DMI 6000 microscope, cell migration was captured over 24 hours at 30 min intervals. Using ImageJ software, the area of the wound was measured, and the percentage closure determined at each timepoint. Triplicate experiments were analysed by two-way ANOVA.

## Cell proliferation and clonogenic assays

Proliferation assays were performed in 96-well plates with cells seeded at $1\times10^3$ per well. Cells were treated with drug or vehicle (DMSO) for 5 days at the indicated concentrations, and metabolic rate was assessed using alamarBlue (Thermo Fisher Scientific). IC50 values were determined using GraphPad Prism version 8.

Clonogenic assays were performed in 12 well plates at $5\times10^3$ cells per well, with 100 μM R-ketorolac, 25 μM NSC23766, 6.25 μM 1A-116 or DMSO. Drugs were replenished every 2–3 days for 1 week. Colonies were fixed with 16% v/v trichloroacetic acid and stained with 10% v/v Diff Quik Stain 2 (Lab Aids Pty Ltd). Plates were scanned using Epson Perfection V800 photo scanner at 1200 DPI. Colony area was quantified using ImageJ software.

## Transwell migration assay

Cells were serum starved overnight, then detached using 5 mM EDTA in PBS. Cells were plated in duplicate at a density of $4\times10^4$ cells per well into the top chamber of a Transwell in serum-free media. Cells were allowed to migrate towards serum-containing media in the bottom chamber of the Transwell for 24 hours at 37 °C. Cells that had not migrated were removed using a cotton bud, before migrated cells were fixed for 10 seconds and stained using the DiffQuick Staining Kit.

Migrated cells were imaged using an Olympus CKX41 light microscope, 5 random fields of view were imaged per well. Images were analysed using ImageJ and migrated cells were counted using the "Cell Counter" plugin.

## Flow cytometry cell cycle analysis

Cells stained with 10 μg/mL propidium iodide (Sigma-Aldrich) for 2–5 hours and incubated with 0.5 mg/mL RNase A (Sigma-Aldrich) were analysed on a FACS Canto II. Data were analysed using FlowJo (BD Biosciences) on at least 20,000 events.

## Intraductal xenografts

Intraductal xenografts were performed as described in[90]. In brief, immunocompromised NOD-SCID-IL2Rγ -/- (NSG) mice were housed in individually ventilated cages and specific pathogen free conditions in a 12 hour:12 hour light:dark cycle with food and water given *ad libitum*. After an acclimatisation period of a minimum of seven days, intraductal injections were modified from a previously described protocol without a Y incision in the abdomen[91]. $8\times10^4$ Endocrine Tolerant or Fast-growing cells were unilaterally injected into the nipple of the 4th mammary gland of each mouse, aged 6-8 weeks old, with 10 mice per cohort. Tumour growth was assessed weekly using callipers for up to 9 months, until 10% tumour burden was reached, or prior if there were major clinical signs. Maximum tumour size of 10% body weight was not exceeded in experiments. Intraductal tumours are shaped as flattened oblongs rather than ovoids, leading to a tumour volume estimate of (length x width x width/2)/2. Mice were euthanised with Isoflurane anaesthetisation and cervical dislocation and the primary tumours, lungs and gynaecological organs were harvested and fixed for 24 hours in 10% buffered formalin at room temperature. All animal procedures were approved by the Garvan/St Vincent's Animal Ethics Committee (Animal ethics number 17/23).

## Hematoxylin/eosin (H&E) staining and collagen/SMA detection in tumour tissue

Paraffin embedded mammary tumours were sectioned, deparaffinized, rehydrated and stained with hematoxylin and eosin (Australian Biostain). Qupath software[92] was used to analyse H&E stained sections of mammary gland tumours, and to detect nuclei. Pixel size of nuclei was used to demarcate nuclei into small (> 10- < 50 pixels) and large (> 50 pixels). Tumours were further stained with α-SMA (1:100, #ab5694, Abcam) using the Bond RX Automated Stainer (Leica Biosystems). Heat induced epitope retrieval was performed at pH 9 (Bond Epitope Retrieval solution 2, Leica Biosystems), 100 °C for 30 mins. Detection was performed with diaminobenzidine and slides were counterstained with haematoxylin. Slides were imaged using a slide scanner (AperioCS2, Leica Biosystems), and images were analysed using QuPath[92].

SHG signal of the extracellular matrix was performed as in ref. 93. SHG was detected using a 25 × 0.95 numerical aperture water objective on an inverted Leica DMI 6000 SP8 confocal microscope. Excitation source was a Ti:Sapphire femtosecond laser (Coherent Chameleon Ultra II) at 80 MHz and tuned to a wavelength of 880 nm and intensity was recorded using an RLD-HyD detector (440/20 nm band pass emission filter). 10 μm H&E sections were imaged at 1.706 μm optical sections. 4 representative fields of view (512 ×512 pixel) per condition were imaged and the intensity of SHG signal was quantified using ImageJ software. Representative images show maximum intensity projections.

To analyse collagen by picrosirius red, 4 μm sections of fixed samples were deparaffinised, rehydrated, and stained with 0.1% picrosirius red (PR) (Polysciences) for fibrillar collagen according to manufacturer's instructions. Once stained they were digitised by the Olympus Slideview VS200 using a 20X objective. The slides were scanned in fluorescence mode with the Cy3 (Excitation: 554/23,

Emission: 595/31) filter at 70% intensity with 50 ms exposure time and high focus settings. Quantitative area measurement of fibrillar collagen PR-Cy3 signal was carried out using ImageJ/FIJI and the relative area (as a % of total tumour area) was then calculated. The ImageJ/FIJI script is available via GitHub (https://github.com/TCox-Lab).

### Characterisation of metastases from intraductal xenografts to multiple organs and gynaecological abnormalities

Metastases were identified in lung sections by IHC for human cytokeratin. Mouse lungs were sectioned and stained with Cytokeratin (1:200, #MA1-12594, Invitrogen) using the Bond RX Automated Stainer (Leica Biosystems). Heat induced epitope retrieval was performed at pH 9 (Bond Epitope Retrieval solution 2, Leica Biosystems), 100 °C for 30 min. Detection was performed with diaminobenzidine and slides were counterstained with haematoxylin. Slides were imaged using a slide scanner (AperioCS2, Leica Biosystems), and images were analysed using QuPath[92] by annotating and measuring the area of the stained regions.

A subset of animals was analysed for cancer cells in the bone marrow. This analysis was limited to a subset of animals as this endpoint was added during the experiment. Bone marrow from the femur and tibia of mice were flushed with PBS, additional cells collected from crushed bone, and passed through a mesh filter. Cell pellets were collected by centrifugation at 1200 rpm, for 5 min at 4 °C and resuspended in 200 μL of PBS, prior to filtration with a 40 μM filter. Cells were blocked with 1 μL of FC block (BD Biosciences) for 10 min on ice followed by incubation with PE human CD298 (1:500, #341704, BioLegend) for 20 min. Cells were resuspended in 500 μL of FACS buffer (2% FBS in PBS), and co-stained with DAPI (0.5 μg/mL, Invitrogen). Cells were analysed on FACS Canto II (BD Biosciences).

Gynaecological organs were scored for abnormalities upon autopsy. Abnormalities included vestigial male organs, uterine hyperplasia, white spots on ovaries and haemorrhagic ovarian cyst. Representative images were collected using an iPhone 8.

### MMTV-PyMT model

Rac1-FRET biosensor mice were crossed to the MMTV-PyMT mice, generating a spontaneous mammary tumour model (MMTV-PyMT/Rac1-FRET) with detectable Rac1 activity[42]. Mice were implanted with 60 day release 5 mg tamoxifen pellets (Innovative Research of America) upon detection of palpable tumours. Mammary tumours were measured thrice weekly using callipers and endpoint was a cumulative tumour burden of 8–10% of body weight. Maximum tumour size of 10% body weight was not exceeded in experiments. Mice with an average mammary tumour size of 120 mm³ were implanted with a mammary window to image Rac1 activity[42], with all procedures approved by the Garvan/St Vincent's Animal Ethics Committee (Animal ethics number 19/13).

Mammary tumours of MMTV-PyMT/Rac1-FRET biosensor mice were collected to derive a cell line. Tumours were minced and cells cultured in 10% FBS, DMEM, 1% penicillin/streptomycin, 5 μg/mL insulin, 10 ng/mL EGF, and 10 ng/mL Cholera Toxin A. A chronically treated tamoxifen cell line was derived with 1 μM 4-hydroxytamoxifen (Sigma-Aldrich; resuspended in tetrahydrofuran (THF)) for ~30 passages until cells were constantly proliferating. Matched control cells treated with vehicle (THF) were cultured for a similar number of passages prior to experimentation.

### PDX model

PDX KCC4653 was generated under the Human Research Ethics Committee (HREC)-approved protocol at the St Vincent's Hospital (SVH) (Protocol HREC/16/SVH/29) and in vivo experiments were approved by the Garvan Animal Ethics Committee (Protocol 24/16 and 21/09). This PDX was established from a metastatic ER + PR+ liver biopsy obtained from a patient with ER+ breast cancer who had disease progression on adriamycin and cyclophosphamide, tamoxifen, letrozole, denosumab, exemestane, capecitabine, methotrexate and abemaciclib. Patient tumour tissues were implanted into the 4th inguinal mammary gland of 6-8 week old female NOD-SCID-IL2γR − /− (NSG) mice (Australian BioResources) as previously described[94]. Animals were anesthetised with isoflurane, and analgesia implemented with ketoprofen and bupivacaine. Mice were subcutaneously implanted with an estradiol pellet (0.4 mg) at surgery, and subsequently supplemented with estradiol in their drinking water for one month post-surgery (8 μg/mL) to support tumour growth.

For therapeutic experiments, a single PDX tumour was divided and implanted into a cohort of 7-week-old female NSG recipient mice. Tumour-bearing mice were randomised to treatment arms when tumours reached 120–150 mm³: vehicle (45% saline/45% PEG300/ 5% Tween 80/5% DMSO administered by oral gavage 5 days a week), 4 mg/kg NSC23766 (SelleckChem) in water administered by intraperitoneal injection 5 days a week, 10 mg/kg R-ketorolac ((MedChemExpress) in 45% saline/45% PEG300/ 5% Tween 80/5% DMSO administered by oral gavage 5 days a week), tamoxifen ((Innovative Research America) 60-day release, 5 mg/pellet implant) or combinations of tamoxifen/NSC23766 or tamoxifen/R-ketorolac. Mice were supplemented with estradiol (2.5 μg/mL) in drinking water to support tumour growth.

Tumour volumes were assessed twice weekly by calliper measurement and calculated using the formula (length × width ×width)/2. Mice were euthanised when the tumours reached endpoint of ~1000 mm³, or prior if mice presented with major adverse clinical signs.

### Toxicity assessments

Peripheral blood was collected by tail vein bleed from NSG mice implanted with PDX and anticoagulated using 10% EDTA in water in order to assess toxicity of treatments. Bloods were collected at the beginning of treatment (0 weeks), and then at 3, 6, 9, 12, and 15 weeks following commencement of treatment. Samples were analysed using an Abaxis VetScan HM5 (Zoetis) haematology analyser. All mice were bled and analysed at each timepoint; however, some data points were missing due to technical failure of those samples during processing.

### DropSeq capture method, library preparation and sequencing

Cultured cells were captured using DropSeq microfluidic devices as described in ref. 95 with modifications and adaptations described in ref. 96. Short-term treatment samples were collected and sequenced in a single batch to minimise batch effects. In brief, dissociated cells were mixed with DropSeq barcoded beads and droplet generation oil to generate droplets containing single cells and beads. This was performed using syringes with set flow rates on a PDMS co-flow microfluidic droplet generation chip (FlowJem). Droplet captures were performed for 7 min to obtain 0.5 mL of beads, with ~2.5% of beads associated with a cell. Following droplet lysis, beads with barcoded RNAs were retrotranscribed, treated with exonuclease I, and amplified by PCR. Pooled cDNAs were then tagmented with the Nextera XT DNA Library kit (Illumina Cat# FC-131-1024) and custom primers as in ref. 96. Tagmented and multiplexed cDNA libraries were sequenced as in ref. 96 on a Nextseq 500 using Nextseq 500 High Output v.2 kit (75 cycles, Illumina Cat# FC-404-2005).

The sequencing output was analysed using the McCaroll lab cookbook, a custom genome (hg38 plus Trinity assemblies of transgene sequences[97]) and gene annotation (gencode vM14 plus). Seurat (v Seurat_5.1.0[98]) was the main platform for visualisation and expression analysis. Differential gene expression (DGE) matrices were imported into the R statistical environment and records were trimmed for quality metrics (> 200 genes, <15% mitochondrial reads, ≥1200 UMI and genes expressed in at least 3 cells). Following quality control, the numbers of cells suitable for analysis in each sample were: Asynchronous: 2336; 24 h estrogen treatment: 941; 6 h estrogen treatment:

2066; 48 h fulvestrant treatment: 340; Endocrine tolerant: 659; Fast-growing: 578; Parental: 517.

Monocle (v2[16]) was used to identify genes variably expressed (mean expression >0.05, observed dispersion > fitted) between samples (fulvestrant arrested, 6 h estrogen and 24 h estrogen), which were then used to assemble all cells along a "release from arrest" pseudovector using DDRTree. State specific markers from this analysis were used in conjunction with Cell Cycle scoring (based on database for Cell Cycle: cc.genes.updated.2019[99]) to assign dominant cell cycle transcriptional signatures to cell clusters. Differential gene expression between conditions were identified using the Tobit test (adjusted $p < 0.05$, average log foldchange >1) and functional annotation was performed using the clusterProfiler package in R.

### Gene set enrichment and network analysis

A top list of 288 differentially upregulated genes in each of the Endocrine Tolerant and Fast-growing cells compared to parental cells was identified. This cutoff included all genes ($n = 288$) with an FDR of <0.05 when comparing Endocrine Tolerant cells to parental cells, and the same number of top genes ($n = 288$, FDR < 0.05) with a false discovery rate of <5 when comparing Fast-growing cells to parental cells. GSEA was performed by comparing to the molecular signatures database (Curated, MSigDB)[100,101] with ShinyGo[102], with the top 10 pathways with the lowest false discovery rate identified. Gene lists were further analysed for gene regulatory networks by using "signalling network analysis", the Signor 2.0 database[103] and Steiner Forest network analysis via NetworkAnalyst[27,104].

Nodes identified in pseudotime analysis were also analysed by GSEA. The top 100 differentially upregulated or downregulated genes of Node#1, Node#2 and the remaining cycling cells were selected. These were compared to the molecular signatures database (Curated, Reactome) with ShinyGo, with the top 20 pathways with the lowest FDR identified.

### Mammary imaging windows

In vivo imaging via mammary windows and their insertion into the skin is described in ref. 105,106. Ethics approval for this procedure (ARA 16/13, 19/13) was via the Garvan/St Vincents Animal Ethics Committee. The procedure is described in detail in ref. 42. In brief, cyanoacrylate glue was used to adhere a 12 mm coverslip to the inset of a titanium ring (Russel Symes & Company) 24 h prior to surgery. The titanium ring and window was disinfected with 70% ethanol.

Female mice were administered 5 mg/kg of carprofen analgesic in drinking water from 24 hours prior to and for at least 72 hours following surgery (Rimadyl[107]). At surgery, mice were anaesthetised by initial exposure to 4% isoflurane (in $O_2$), followed by maintenance in 1.5–2% isoflurane in $O_2$, using a calibrated vaporiser. Mouse body temperature was maintained via placement upon a heating pad and eyes lubricated with LacriLube. 100 μL of 0.075 mg/kg buprenorphine for pain management was subcutaneously injected, and again 6 hours after surgery. The incision site was shaved, depilated using hair removal cream (Nair), and disinfected (0.5% chlorhexidine/ 70% ethanol). Using microdissection scissors, an incision was made in the skin directly over a palpable mammary tumour, followed by blunt dissection of the surrounding tissue. A purse-string suture was then threaded through the skin around the opening. The imaging window was integrated by carefully seating the skin edges into its lateral groove. To ensure the window sat flush against the tumour mass, any trapped air was removed via subcutaneous aspiration. Following the procedure, the mice were given 72 h to recover and were transitioned off carprofen at least 24 h before in vivo imaging commenced. Post-surgery mice were supplied with sunflower seeds and/or recovery gel in order to aid recovery. Metal food hoppers and plastic domes were removed from the cages and the feed supplied in food trays on the floor of the cage in order to avoid any damage to the window by the mice and cage

surroundings. Cage enrichment was supplied in the form of paper mache domes as well as tissues to serve as nesting material.

### In vivo and in vitro imaging, single cell FLIM analysis and motion correction

The imaging strategy has been described previously in ref. 42, and is reiterated here. Mice implanted with an optical window were imaged on a heated stage (Digital Pixel, UK) and anaesthetised using 1–2% isoflurane. Imaging was performed on mice bearing tamoxifen tablets, and additionally after i.p. injection of the $H_2O$ vehicle or 4 mg/kg NSC23766, or oral gavage with 1 mg/kg R-ketorolac or vehicle (saline, 45% PEG300, 5% Tween80, <5% DMSO). Images were acquired on an inverted Leica DMI 6000 SP8 confocal microscope setup using a $25 \times 0.95$ NA water immersion objective. The Ti:Sapphire femtosecond laser (Coherent Chameleon Ultra II, Coherent) excitation source operating at 80 MHz was tuned to a wavelength of 840 nm. Collagen I and ECFP intensities as well as the FLIM of ECFP were detected using RLD-HyD detectors with 435/40 nm and 483/40 nm bandpass emission filters respectively.

Images were acquired at a $512 \times 512$ pixel resolution, a line rate of 700 Hz and to a total of 203 frames per image. Galene (v2.0.2[108]) was used to correct for physiological movement during imaging using the warp realignment mode, 10 realignment points, a smoothing radius of 2px and a realignment threshold of 0.4 applied for the SHG channel and 0.6 for the ECFP signal. Single cell analysis of the Rac1 activity was performed using FLIMfit (v5.1.1) by drawing ROIs around the cell membranes of individual cells. Merged images of the FLIM signal and the intensity are shown, with the FLIM heatmaps depicting active Rac1 (high-FRET) as green/blue colours, inactive Rac1 (low-FRET) by red/yellow colours, and black areas representing a lack of signal.

### Analysis of patient samples/PDX cohorts

**Autopsy cohort.** IHC data from a breast cancer autopsy cohort[9] was interrogated to determine the change in Ki67 protein expression between primary and metastatic disease. In brief, the cohort included ER+ and ER- patients with matched primary and metastatic disease that had been previously characterised for ER, PR and Ki67 protein expression. The 19 ER+ patients had 95 matched metastases, and the 38 ER- patients had 163 matched metastases. Ki67 expression was binned into 7 expression categories as follows: 0 = negative, 0.5 = rare cell; 1 = 1%, 2 = 2-10%; 3 = 11-24%; 4 = 25-50%, 5 = > 51%. Each metastasis was compared to the matched primary tumour to determine if Ki67 increased or decreased. Data was compared by paired two-sided t-tests.

**Gynaecological metastasis cohort.** 54 breast cancer patients with gynaecological metastases[46] were assessed for P-Rex1 expression by immunohistochemistry using the Bond RX Automated Stainer (Leica Biosystems). Anti-P-Rex1 Antibody (1:500, #HPA001927, Sigma-Aldrich) was used to stain tissues following heat induced epitope retrieval at pH 6 (Bond Epitope Retrieval solution 1, Leica Biosystems), 100 °C for 30 min. Detection was performed with diaminobenzidine and slides were counterstained with haematoxylin. Kidney tissue was used as a negative control. P-Rex1 staining was assessed blinded by pathologist E.K.A.M. using the H-score system of % positive and stain intensity[109]. High P-Rex1 was defined as H-score >50, with scatter plot of data distribution shown in Supplementary Fig. 6E. A.M.R. performed survival analyses and correlated scores to previously identified ER+ status and site of metastases[46]. Survival analyses were performed using Log-rank (Mantel-Cox) tests on GraphPad Prism. Hazard ratios (Log-rank) were computed for each analysis, and reported along with the 95% confidence interval. A subset of the cohort has matched primary and metastatic biopsies. Matched primary and metastatic pairs were separated into groups where ER was increased/sustained, or decreased. Note that in these analyses some primary tumours were

matched to multiple different metastases. P-Rex1 expression was compared within these groups to determine if P-Rex1 was significantly different, using paired two-sided t-tests.

**scRNAseq of breast cancer autopsy samples.** Samples were collected from a patient who had received systemic treatment with tamoxifen, zoladex, aromatase inhibitors, denosumab, capecitabine and eribulin. Samples were collected through the MonARC Programme, approved human ethics protocol SVH17/173 with informed consent, and analysed under the Personalised Medicine for Breast Cancer study, protocol x19-0496. Normal and tumour single-cell suspensions from patient's lungs were prepared, captured on 10X Genomics platform and sequenced on Illumina NextSeq 500 as described[110]. Raw sequencing data were demultiplexed and aligned to GRCh38 genome with Cell Ranger software (version 3.1.0), normalised with Seurat (v.3.1.2) in R (v.3.6.1), and visualised with Loupe Browser 5.1.0.

**PDX samples.** Formalin fixed paraffin embedded samples of PDX tumours were screened for P-Rex1 expression by immunohistochemistry using the staining protocol described above. PDX are listed in Supplementary Table 1 and described in refs. 111–113, and KCC4653 is described above.

### In silico datasets

**Single cell primary breast cancer dataset.** Data on 24,489 epithelial cancer cells from 26 primary breast cancers[114] including 11 ER + , 5 HER2+ and 10 TNBCs was accessed using the Broad Single Cell Portal (https://singlecell.broadinstitute.org/single_cell) to interrogate the relationship between gene expression and single cell identity in different epithelial lineages, and data is available at GSE176078.

**Survival analyses of endocrine therapy treated patients.** Normalised composite datasets of relapse free survival of endocrine therapy treated patients were accessed via KMplotter[115]. Probes for *PREX1* (224909_s_at), *MKI67* (212023_s_at), *EGFR* (211551_at), *PAK2* (208875_s_at) and *RAC1* (208640_at) were analysed. Patients treated with endocrine therapy were demarcated into high and low expression of each gene based on median score. Survival analyses were performed using Log-rank (Mantel-Cox) tests on GraphPad Prism. Hazard ratios (Log-rank) were computed for each analysis.

**TCGA, METABRIC, Metastatic Breast Cancer Project and CPTAC datasets.** RNA and RPPA expression data from TCGA primary breast cancers[116,117], Metabric primary breast cancers[118] and Metastatic Breast Cancer Project (MBCP[40]) were accessed via cBioPortal[119,120] in May 2022. ER+ cancers were defined using the Metabric expression based classifier, by IHC classification for TCGA, or pathology (PATH) reporting for MBCP. Protein expression data for P-Rex1, Ki67, EGFR, Pak2 and Rac1 from 18 normal breast tissues and 90 primary breast cancers were analysed from the Clinical Proteomic Tumour Analysis Consortium (CPTAC)[121] dataset. Data and statistical comparisons between breast cancer subtypes were accessed via UALCAN[122].

**PyMT-MMTV primary, disseminating and metastatic cancer cells.** Microarray analysis of residual disease, disseminating tumour cells, and metastases in the MMTV-PyMT breast cancer model (GSE43566[43]) was accessed using the NCBI GEO archive[123]. Extracted data was analysed by GraphPad Prism software.

**Genes deregulated in breast cancer dormancy.** List of genes identified as differentially upregulated in dormancy were downloaded from[26]. Gene list was sorted for those upregulated in dormant cells, and *Mus musculus* gene identifiers were converted into human

identifiers using g:Profiler[124], to create a human orthologs list for comparison with other datasets.

### Meta-analysis

Publications reporting peri-operative analgesic usage and breast cancer recurrence were identified through searches with the terms "analgesic", "NSAID", "ketorolac" and "breast cancer recurrence" of the PubMed database and Google Scholar. To be included, studies needed to report disease-free survival or recurrence-free survival following peri-operative analgesic treatment, and selected studies are listed in Supplementary Table 2. Studies examining systemic long-term use were excluded. Studies on ketorolac included two studies where a small proportion of patients had been treated with diclofenac, as indicated in Supplementary Table 2. Bias within each meta-analysis was assessed by inspection of FUNNEL plots (Supplementary Fig. S10A, B) of treatment effects from individual studies plotted against a measure of study size[125]. A random effects model was used due to the high heterogeneity between the studies. Meta-analysis was performed using Revman 5.0 software[126], and hazard ratios and confidence intervals were re-generated by Revman 5.0 once raw values were entered in the software. Forest plots were generated with Revman 5.0.

### Statistics and reproducibility

Experiments were performed in biological triplicate, except where indicated. Data were analysed in GraphPad Prism, using t-tests, one-way ANOVA, two-sided non-parametric Mann–Whitney tests, two-way ANOVA or nested t-tests, as appropriate. When one-way ANOVA was used, the differences between individual samples were compared using Tukey's multiple comparisons test, or Dunnet's multiple comparison test. Where two-way ANOVA was used, multiple comparisons were performed with Bonferroni's multiple comparison test, or as appropriate. For proportional analyses, a contingency analysis with a chi-squared test was performed. A p-value of <0.05 was deemed significant throughout. Error bars in each figure are as indicated in the figure legends.

No statistical method was used to predetermine sample size, but generally 3–12 replicates were performed for in vitro experiments, and 4–12 animals per arm in the animal experiments. No data were excluded from the analyses. Allocation of animals in animal studies was randomised. Representative subsets of 5–8 tumours were analysed per arm in Fig. 1J, L, Supplementary Fig. 1H, with no data excluded. One animal was not analysed in Fig. 1M, O as tissue could not be collected. In the IHC analysis shown in Fig. 5H–L, the pathologist (E.K.A.M.) was blinded to the identity of the samples, and the investigator (A.M-R.) was blinded to expected outcome.

### Reporting summary

Further information on research design is available in the Nature Portfolio Reporting Summary linked to this article.

## Data availability

Single cell datasets relating to Fig. 2 have been deposited in GEO Datasets at GSE306192. The Fig. 5 scRNAseq data generated in this study have been deposited in the EGA database under study accession EGAS00001008353. EGAS00001008353 data are available under restricted access for reasons of patient confidentiality, access can be obtained through application to the Data Access Committee. Processed datasets associated with EGAS00001008353 are available at GEO datasets at GSE306192. The datasets GSE43566 and GSE176078 which were reanalysed for this study are available at GEO datasets. The data portals KM plotter [https://kmplot.com/analysis], UALCAN and cBioPortal [https://www.cbioportal.org] were used to access other listed publicly available datasets. Otherwise, the data supporting the findings of this study are available within the paper,

or in its Supplementary Information. Source data are provided with this paper.

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

## Acknowledgements

We express our gratitude to the patients and their families who generously donated tissue that was analysed in this project. We also extend our thanks to consumer advocates who were involved in the development of this project, particularly Vanessa Smith. We acknowledge the Histopathology Facility, Flow Cytometry Facility, Australian BioResource (ABR), the Biological Testing Facility, the Tissue Culture Facility under Gillian Lehrbach and Ruth Lyons, and the ARCF INCITe Centre at the Garvan Institute, Sydney, Australia for their services. We thank A/Prof David Croucher, Dr Aurélie Cazet and Dr Niall Byrne for advice on experimental techniques. We thank the Swarbrick lab for development of the PDX model, especially Jessica Yang. We thank John Reeves for assistance with data uploads, and Dr Zoe Phan for reading the manuscript. This research project was supported by the Australian National Health and Medical Research Council (NHMRC) APP1138077 and NBCF grant IIRS21-066. Cohort development was supported by programme funding to S.R.L. from NHMRC (APP1113867). E.L. was supported by a National Breast Cancer Foundation Endowed Chair (EC17-02). M.N. was supported by a NHMRC Ideas grant (APP1188208), a CINSW (ECF171227) and St Vincent's Foundation Thelma Greig Cancer Grant. T.R.C., A.S., P.T., and C.M. are supported by NHMRC Investigator grants (2033065, 1161216, 2016930, 2016951). E.T.Y.M is supported by an Australian Government Research Training Programme Scholarship and Tour de Cure. C.E.C. was supported by a National Breast Cancer Foundation Career Development Fellowship (ECF17-002), Cancer Institute NSW Fellowship (2020/CDF1071) and the Lysia O'Keefe Fellowship. A.S. was supported by the generosity of J. McMurtrie (AM), D. McMurtrie and the Petre Foundation, and is a Breast Cancer Research Foundation (BCRF) investigator.

## Author contributions

B.G. and L.E. contributed equally to the manuscript. This study was conceived and designed by C.E.C. K.J.F., G.S., M.N., and C.E.C. designed experiments, collected, analysed, and interpreted data. Intraductal animal experiments were undertaken by S.R.O., K.J.F., and C.L., and analysed by K.J.F., and G.S. In vivo imaging experiments and associated models were developed by M.N. and P.T., and performed by V.L., J.S., K.J.F., G.S., and M.N. Additional in vitro experimentation was performed by C.L. and S.A. E.J. performed shRNA experiments and E.J., L.O., and C.M. analysed data. A.M-R., J.R.K., M.C.C., P.T.S., and S.R.L. collated and annotated patient cohorts, E.K.A.M. scored cohorts and A.M-R. performed blinded analysis of the data. Single cell data generation of models was performed by K.J.F., D.G-O., F.V-M., Y.C-O., R.S., and analysed by L.E. and B.G., with contributions from D.R., and C.E.C. L.E. and B.G. designed and performed integrative data analysis and contributed to overall single cell data interpretation. Additional patient samples for scRNAseq were sourced and analysed by E.L., S.W., K.H., N.B., A.S., and C.E.C. PDX model was derived by the lab of A.S., experimentation performed by K.J.F., M.H.S., and C.L., with data analysis by K.J.F., E.T.Y.M., H.L.W., and T.R.C. C.E.C. drafted the manuscript with input from all authors, and particularly K.J.F., S.A., A.M-R., P.T., and M.N. All authors read and approved the final manuscript.

## Competing interests

The authors declare competing interests.

## Additional information

¹Garvan Institute of Medical Research, Sydney, NSW, Australia. ²School of Clinical Medicine, Faculty of Medicine and Health, UNSW Sydney, Sydney, NSW, Australia. ³Division of Cell Matrix Biology and Regenerative Medicine, Manchester Cell-Matrix Centre, School of Biological Sciences, Faculty of Biology Medicine and Health, The University of Manchester, Manchester, UK. ⁴Westmead Research Hub, Westmead Institute for Medical Research, Westmead, NSW, Australia. ⁵The University of Queensland, Faculty of Health, Medicine and Behavioural Sciences, Centre for Clinical Research, Brisbane, QLD, Australia. ⁶Cancer Program, Monash Biomedicine Discovery Institute and Department of Biochemistry and Molecular Biology, Monash University, Melbourne, VIC, Australia. ⁷St George and Sutherland Clinical Campuses, School of Clinical Medicine, UNSW Medicine and Health, UNSW Sydney, Sydney, NSW, Australia. ⁸Department of Anatomical Pathology, NSW Health Pathology, Kogarah, NSW, Australia. ⁹ANZUP Cancer Trials Group, Health Translation Hub, Randwick, Sydney, NSW, Australia. ¹⁰The George Institute for Global Health, Health Translation Hub, Randwick, Sydney, NSW, Australia. ¹¹Cancer Epigenetic Biology and Therapeutics, Therapeutic Discovery Theme, Children's Cancer Institute, Sydney, NSW, Australia. ¹²Institute for Tissue Medicine and Pathology, University of Bern, Bern, Switzerland. ¹³Pathology Queensland, The Royal Brisbane and Women's Hospital, Brisbane, QLD, Australia. ¹⁴Institute for Biomedical Materials and Devices, University of Technology Sydney, Sydney, NSW, Australia. ¹⁵School of Biomedical Engineering, Faculty of Engineering and IT, University of Technology Sydney, Ultimo, NSW, Australia. ¹⁶St Vincent's Hospital, Darlinghurst, Sydney, NSW, Australia. ¹⁷These authors contributed equally: Kristine J. Fernandez, Ghazal Sultani. ✉e-mail: l.caldon@garvan.org.au

