## [Transparent Peer Review file · Nature Communications]

Endocrine therapy reprogramming of breast cancer facilitates metastatic escape via upregulation of P-Rex1/Rac1 signalling

Corresponding Author: Dr C. Caldon

Version 0:

Reviewer comments:

Reviewer #1

(Remarks to the Author)

please see attached PDF for my comments

Reviewer #2

(Remarks to the Author)

Summary: In this manuscript, the authors identified a novel prognostic biomarker predicting late recurrence in estrogen receptor (ER)-positive breast cancer. They first sorted endocrine therapy resistant MCF-7 ER+ breast cancer cells into a fast and slow growing population and characterized their cellular characteristics and metastatic potential using in vivo xenograft models. Using single-cell RNA sequencing (scRNA-seq) analysis, they found that RAC1 signaling is upregulated in the slow-growing population compared to the fast-growing population. In addition, they showed that the expression of P-Rex1 (RAC1 signaling component) is increased in ER+ metastases and is associated with late recurrence.

The present manuscript is potentially important because it proposes a novel prognostic biomarker and therapeutic target for late recurrence in ER+ breast cancer. However, there are several major points that should be addressed to substantiate the authors' claims.

Major points:

1. Figure 2A: The authors concluded that resistant populations do not arise from pre-existing drug-tolerant persistent (DTP) cells due to the lack of obvious overlap between the parent and resistant cells. However, this approach is too ambiguous and subjective to support this argument. In addition, we cannot exclude the possibility that DTP cells are induced and activate a transcriptional program that confers a selective advantage upon drug treatment. To rigorously address this issue, the authors should use a cellular barcoding system to track the lineage of cells over time.
2. Supplementary Figure 2A and 2B: The estimated cell cycle phase by DNA content (Supplementary Figure 2A) and scRNA-seq (Supplementary Figure 2B) is not consistent. This issue should be carefully investigated, as the main results from scRNA-seq (Figure 2) are highly dependent on the cell cycle phase predictions.
3. Figure 2D and 2E: In the pseudotime analysis, it is unclear what the biological meanings of the inferred pseudotime are in this context. How robust are the inferred trajectories across replicates and different inference algorithms?
4. Figure 2G: The upregulation of MT-ATP8 may be explained by the increased percentage of reads mapped to mitochondrial genes, a key technical feature of low-quality cells.
5. Figure 2I and 2L: Line 318-320 on page 15 is unclear. What does "its substantial clustering with Arrest #2" mean? The beta-galactosidase level of the "Fast growing resistant" population was more similar to that of "48h Fulv" compared to the "Endocrine tolerant" population in Figure 2L, which is not consistent with Figure 2I?
6. Figure 6F-H: The authors showed that RAC1 inhibitors reduce the RAC1 activity in vivo. However, this is not surprising. To demonstrate the efficacy of the RAC1 inhibitors, they should show that these drugs have a synergistic effect on reducing cancer cell survival and drug resistance with endocrine therapies in vivo.

Minor points:

1. "Figure 1O" at line 256 on page 12 and "Fig 1P" at line 259 on page 12 should be replaced with "Figure 1N" and "Figure 1O", respectively.
2. Figure 2G, 3I: The statistical significance should be provided.
3. Figure 2J: The resolution of the heatmap should be improved. What are the common upregulated genes between "48h fulv" and "Tolerant" and their shared biological characteristics?
4. Many supplementary figures (e.g. Supplementary Figure 2C and 3H) were not referenced in the main text, or were referenced inaccurately, and were unedited. It appears that the manuscript was put together in a rush.
5. Figure 3B and 3C: What do the node colors mean?
6. "Supp Fig S3H" at line 374 on page 17 should read "Supplementary Figure 3I". Then, Supplementary Figure 3H is not referenced.
7. Supplementary Figure 4: No labels. Supplementary Figure 4A referenced on line 400-401 on page 18 does not exist.
8. Figure 4I: The signal activity should be quantitatively compared.
9. Figure 5E and Supplementary Figure 5C: In the axis labels, "tSCNE" should be replaced with "t-SNE".

Reviewer #3

(Remarks to the Author)

This work investigates the properties of slow proliferating ER+ breast cancer cells that can withstand hormone therapies. The authors observe that these have many different properties from fast growing resistant cells, notably low EGFR signaling and high Rac1 activity, possibly linked to Prex1. Interestingly, they are at least as efficient as fast-growing cells in their ability to metastasize. Through a neat series of experiments and analysis, the authors demonstrate that hormone therapy selects for high Rac1 activity in the MMTV-PyMT model and potentially in patients. They conclude by showing the potential utility of targeting Rac1 pharmacologically. This is done using the widely used NSC23766 compound and an enantiomer of the analgesic ketorolac. Intriguingly, analysis is presented to suggest that ketorolac has anti-metastatic properties not observed in other NSAID analgesics. There is much to like about this work; it is interesting, novel, well-written, and broadly convincing study. However, there are some notable gaps and loose ends that need addressing before publication.

Specific comments

1. There is an interesting change in the collagen network in the slow tolerant MCF7 cells (Fig1j). Is this observed in the PyMT model – SHG images should be shown? Does the transcriptomic data provide possible explanation for the molecular basis? Is Rac1 activity driving the altered collagen (this can be determined in experiments suggested below)? Do the tumors have more CAFs – α SMA staining or similar would be informative?
2. Do the authors observe similar fast and slow populations when selecting the resistant PyMT cells? This analysis is required to connect different parts of the paper.
3. Why are the Prex1 high and low groups asymmetric in the Kaplan Meier analysis (Fig 5j). They should really be split into equivalent size high and low groups or tertiles. The asymmetric split suggests some selective choosing of an optimal cut-off. Can the authors either justify their cut-off or re-plot the analysis as suggested?
4. The authors should attempt staining therapy resistant patient samples with a Rac1-GTP antibody and EGFR (or another fast resistant marker).
5. The Rac1 dependence of the slow tolerant cells is never demonstrated in vivo. This is critical and needs to be shown. Ideally, by long-term treatment with NSC23766 or ketorolac, or shRNA/Crispr KO of Rac1. These tumors should be analyzed for collagen SHG (c.f. point #1)
6. Is Prex1 actually required for the growth and metastasis of the slow tolerant cells? Or is it that they are simply Rac1 dependent. The former point needs to be tested experimentally using shRNA/Crispr KO.
7. What are the cohort sizes in the analysis of ketorolac and other NSAIDs?
8. The authors don't discuss the topic of cancer dormancy very directly, but their study clearly relates to it. Some greater discussion is warranted and it would be interesting to cross-reference the transcriptome of the slow tolerant cells with signatures of dormancy.

Version 1:

Reviewer comments:

Reviewer #1

(Remarks to the Author)

The authors have made many modifications of the manuscript in response to reviewer comments. In addition, they have also added important experimental support for their hypothesis, all of which have materially improved this study. There are still some additional concerns about this manuscript, but they require only text changes and edits, not additional experimental work

Concerns:

For figure 1M, the non-significant result for pulmonary metastases appears to be driven primarily by one sample in the endocrine tolerant group. While I applaud the authors for including all of the data, I wonder whether it is worth also analyzing the data with this animal censored. This shouldn't change the overall interpretation of the manuscript, but might better describe what is observed.

Figure 1N, the authors state that the bone mets were "detectable in a subset" of samples. How were these subsets selected? Or are they just looking at those animals with detectable bone mets? Please clarify.

Figure 1O, the authors state that “all mice” showed evidence of gyn abnormalities, but the histogram shows only 8 of the 10 tolerant animals having these abnormalities. Please clarify.

Are the number of epithelial cells sequenced in scRNA seq provided somewhere? I may have missed this but it would be worthwhile reporting

Figures 3L, the Y-axis state normalization to b-actin, but the blot in 3K, as well as the uncropped blot in the supplemental data (fig. S4D) is labelled GAPDH. The band in the uncropped blot also is at the appropriate size for GAPDH. Please correct the figure and associated text.

For the record, in shRNA experiments, it is usually better to analyze more than a single shRNA to avoid off-target effects. However, since the rest of the data is internally consistent, and considering the length of the review cycle for this paper, in this instance I am not suggesting any further delay to perform this additional work.

Please provide a y-axis label for figures 5I and 5J

Figure 7C, for the legend, please move the tamoxifen only treatment to the Single-arm part of the legend rather than the Combination, for clarity.

Reviewer #2

(Remarks to the Author)

The authors have satisfactorily addressed my comments.

Reviewer #3

(Remarks to the Author)

The authors have made reasonable attempts to address the concerns raised or explained why they are not able to do so (e.g. Rac-GTP antibody issues). I recommend publication. My only query is the p value for Figure 5k - is there a typo, should it be <0.0149? If it is <0.149, then this is a rather odd way of indicating no significance.

We thank the reviewers for their positive comments and interest in our study. We were very happy to receive feedback that the reviewers perceived that the study was “interesting, noveland broadly convincing” (R3). In particular it was noted that the “pathway may be important in metastatic progression in ER+ patients” as “ it proposes a novel prognostic biomarker and therapeutic target for late recurrence in ER+ breast cancer” (R2) and it “has the potential of providing an avenue for eventual clinical intervention which might significantly improve survival” (R1).

We have responded to all queries and performed extensive revisions to address the comments of the reviewers. Our major amendments include:

1. We have **completely reanalyzed our scRNAseq data with updated pipelines**, and identified the source of the discrepancy between annotation of cells via scRNAseq signatures versus annotation of the populations by DNA staining. The discrepancy was due to the induction of an intense cell cycle signature as the cells exit cell cycle arrest. We have moderated our results to take this into account, and we have also addressed all additional points regarding the scRNAseq, including using non-mitochondrial genes as an output, and improving the heatmap resolution.
2. We have carefully reviewed all western blots and controls to ensure that there are no discrepancies.
3. We have performed **an extensive in vivo study of RAC inhibitors and endocrine therapy in combination**. For this we have used a novel patient-derived xenograft model from a breast cancer patient who was resistant to multiple lines of therapy. Our work demonstrates that **Rac inhibitors in combination with endocrine therapy are able to induce tumour regression, unlike any of the therapies used as a single agent**.
4. We have systematically addressed every other comment of the reviewers. This includes performing extra staining for SMA, collagen and Ki67 on tumour samples, performing shRNA studies, comparing our datasets to publicly available dormancy datasets, and performing new statistical analyses across multiple datasets.

We apologise to the editorial team and the reviewers for the length of time taken to return the manuscript. We valued all comments and it has taken a substantial amount of time to address these. The *in vivo* studies took a particularly long time: our slow growing PDX model took 6 months to expand from frozen for *in vivo* studies, and then took 8 months to complete, which is commensurate with the slow growing nature of ER+ breast cancers. We also had a serious delay as we were unable to draw conclusions from our first large scale animal study due to systemic contamination issues in the breeding colony that supplies our animal facility, a fact that became known when the study was near completion. As a result, we had to reinitiate the study, causing a delay of 6+ months.

In our responses below the line and page numbers refer to the document without tracked changes. A tracked changes version has also been supplied.

Reviewer #1 (Remarks to the Author)

This manuscript describes the investigation into endocrine therapy and late relapse for ER+ breast cancers. The authors utilize a drug selection method to isolate fast or slow growing cells resistant to endocrine therapy to try to mimic in an experimental setting the important phenomenon of late ER+ breast cancer relapse. Subsequently they perform a variety of in vivo, cellular and molecular analyses to investigate whether or not the slow growing, “Endocrine Tolerant” cells might be responsible for late metastatic relapse. They found that these cells were capable of forming primary tumors, but these tended grow slowly in vivo as in vitro, but had the same metastatic incidence as their fast growing complements. The authors then characterized these cells and determined that the Endocrine Tolerant cells had acquired some characteristics of senescent cells. Subsequently the authors performed scRNAseq to further investigate the molecular cause of these changes and implicated the Rac1 pathway molecule P-Rex1. Further investigations using a Rac1 biosensor and Rac1 pathway inhibitors suggest that this pathway may be important in metastatic progression in ER+ patients. If true, this has the potential of providing an avenue for eventual clinical intervention in this subset of patients which might significantly improve survival for the largest subtype of breast cancer.

Supplemental figure S1E, line 226, tolerant cells look like they are expressing less ER compared to fast cells when comparing relative ratio of ER to b-actin. Densitometry analysis would be helpful to quantitate the differences in ER between the two cell populations. This doesn't change the difference in growth rate between the two types of cells under fulvestrant or tamoxifen, but might explain the difference in growth in the presence of estradiol.

Thankyou to the reviewer for this suggestion. We have performed densitometry, and we found that overall there was no significant difference between the cell lines. These additional data have now been included in Supplementary Figure S1G.

While we did not see a significant difference, the reviewer has made an excellent point, and it may be that other co-factors in the estrogen response pathway are altered between slow growing and endocrine tolerant cells. We re-examined our scRNAseq data to look for these changes, but we did not see any overt differences. However future analysis by conventional RNAseq could provide sufficient sequencing depth to see more subtle changes in estrogen signalling pathways between slow growing and endocrine tolerant cells.

Figure 1M, are the average number of metastases observed in each sample significantly different between the fast and tolerant animals? Incidence and size, once formed, may not be significantly different, but the data suggests a significant reduction in the number of metastases in the slowly growing tumors. This would suggest a decrease efficiency, not an equal efficiency, as stated on line 263.

The reviewer makes an important point in observing that there is a numerical difference in the metastases as there are less overall in the cohort derived from endocrine tolerant cell line. If we collapse all these data to a simple t-test across all animals there is still not a significant difference in the number of metastases between animals with fast growing and endocrine tolerant xenografts ($p < 0.085$).

However, the numerical difference between the models is still a consideration, especially as this could prove significant if we repeated the experiment with a larger animal cohort. For this reason, we have edited the text to be less definitive on the similarity between the two models.

The text now states:

Line 294-295 that “While there were different rates of primary tumour formation from Endocrine Tolerant and Fast-growing cells, these differences were less apparent in metastases.”

And at line 304-306 “Overall, our unique model demonstrates that slow growing Endocrine Tolerant cells form small primary tumours with large nuclei and enhanced collagen deposition, but these cells are capable of metastasising to multiple sites.”

Line 273-4, figure 2A, I disagree. There is overlap between at least the endocrine resistant and the parental population in the center of the graph, which suggests that this population may have pre-existed selection.

As part of the revision process, we used the newest version of Seurat (v5.1.0) to re-analyse the scRNAseq data, as there had been significant advances in this pipeline and package since the data was first analysed in 2018 with Seurat v2.3.4. We also analysed the Endocrine Tolerant, Fast growing and parental populations separately from the other datasets to improve our understanding of overlap between these three populations. From the new analysis we could see that the three populations have now resolved to be more clearly distinct from one another. This is new Figure 2A.

However, based on the comments on Reviewer 1 and Reviewer 2, we agree that this method is not conclusive to draw conclusions about pre-existing populations, and we have moderated our description of the data in the results section. We now state:

Lines 316-323: “Endocrine tolerant cells did not bear a resemblance to one particular subgroup of cells (Figure 2A), in line with a prior study on a model of ER+ aromatase therapy resistance which suggests that endocrine therapy resistance does not arise from particular subclones (30). This does not eliminate the possibility of a persister cell population as we may not have examined sufficient cells, or used the most sensitive method to detect persister cells. However, without a strong indication of persister cells, we hypothesised that differences in proliferation during resistance could alternatively arise from heterogeneity between individual cells in their cell cycle response to endocrine therapy.”

Supplemental figure 2A & B, why are the cell cycle profiles so different? Determining cell cycle state by DNA content should be relatively accurate, with cells exhibiting 3N being considered in S phase. The pattern observed in S2A is consistent with expected re-entry to the cell cycle. Using a transcriptional profile for single cell sequencing, with inherent lack of sequence depth, introduces a potential significant source of error which would significantly affect the remaining data presented in figure 2D-F. The relative cell cycle percentages in S2A make more sense considering the treatments that these cells have undergone. If the subsequent analysis is done using the cell cycle assignments based on DNA content, rather than the transcriptional signature, are the same answers obtained or is the interpretation changed?

Thankyou for this question. Reviewer #2 made a similar point, and we have performed further analysis and amended the manuscript as described under Question 2 of Reviewer #2 questions.

Unfortunately, we are unable to do the analysis in Figure 2 based on DNA content, as we do not have DNA content scoring for individual cells analysed by scRNAseq. We have thus readdressed the discussion of these data with a different focus that does not rely upon cell cycle position for the comparison of the data subsets. This is also described in the response to Q2 of Reviewer #2.

Figure 2K, are these images representative? This does not show a significant difference in b-gal staining. What criteria did the investigators use to call a cell as positive/negative?

Thankyou for this comment. Our original images were at too high magnification and did not show a good representative number of cells. Following on from these comments, we repeated this entire experiment and we have provided new quantitation and new images.

Our criteria for scoring a cell as positive for senescence is to observe β -galactosidase positivity and morphology consistent with a senescent cell. A senescent cell has intense blue staining with accumulation of blue stain in the perinuclear region. These criteria have now been clarified in the Supplementary methods on lines 156-158.

Figure 3G, are the HER2 and pHER2 images from the same blot? The differences in the band patterns suggest that they are not. If this is correct, where is the b-actin control for the pHER2 blot?

We have rerun these samples to ensure that they are sourced from the same blot. New images have been inserted in Figure 3G, and corresponding uncropped western blots in Supplementary Figure 4B.

Figure 3I, the y-axis is listed as "PREX1 relative transcripts", relative to what? Same question for 3J.

We have re-analysed the scRNAseq data in 3H (originally 3I) and inserted new plots showing the different levels of P-Rex1. For these data, the y-axis is labelled log (counts) PREX1.

Other axes within Figure 3 have been amended to no longer state "relative" levels. Figure 3J and 3L now state "Relative P-Rex1/ β -Actin".

GAPDH loading control for the top two panels in figure 3K is not included in the figure. Control is included in figure S3D. This should be included in the primary figure rather than just including the control for the ER blot (S3D right panels), as currently shown. The interpretation that p-Rex1 is upregulated by estrogen supplementation is questionable. If it is true, it is very subtle. Densitometry analysis using the correct GAPDH loading control shown in S3D needs to be performed to substantiate this claim.

Thankyou to the reviewer for their comment and we apologise for not providing the complete dataset in the main figure with our original submission, and for not providing quantitative data. We have now rearranged the figure and included densitometry (Figure 3K, new Figure 3L). The densitometry of this experiment shows that there is a significant increase in P-Rex1 protein with estrogen supplementation. We agree with the reviewer that the differences while significant, are

subtle, which calls for a consideration of their biological relevance. We have thus moderated our conclusion at lines 427-430 to state:

“We also assessed short-term regulation by anti-estrogen and estrogen treatment in parental cells, and P-Rex1 was slightly, but significantly, upregulated by 48 hours of fulvestrant treatment, and then further upregulated by the addition of estrogen (Figure 3K/L, Supplementary Figure 4D).”

Supplemental figure 3C, the Rac1 blot appears to be from a different experiment than the others, based on the curved banding pattern in this blot compared to the others, specifically the faint lower band in the Pak2 blot. The b-actin control for this blot needs to be included in the supplemental figure and added made in figure 3J to reflect that this is from a different experiment.

These bands were all derived from the same experiment and blot. Some commercial polyacrylamide gels contain small imperfections that can cause bands to run unevenly top to bottom, and we believe this has occurred in this case. We have provided a high contrast image below (Figure R1) to demonstrate that the background bands at 25KD in the Rac1 blot are straight, indicating that a lower percentage polyacrylamide patch probably occurred below this region, leading to a small curve through the Rac1 band on lane 5.

UNPUBLISHED DATA FIGURE REDACTED

Line 374, the METABRIC analysis is S3I, not S3H as written. S3F and S3G are the duplicates

Apologies for our poor proofreading: this has been corrected in the text to correspond to the correct new figure numbers.

b-actin blot for figure 3N is not the same as the source blot in S3E. See superimposed image:

We apologise – this was an error on our part. A different exposure was used in the main manuscript to the supplementary data, and the image was inserted upside down when compiling Figure 3N (now Figure 3O). We have now replaced this with the correct image in the main figure.

Figure 4D, what is the “Relative P-Rex1” expression normalized to?

The P-Rex1 protein expression was compared to the matched β -Actin loading control from the P-Rex1 western blots in each case. The figure has been amended so that the y-axis now states “Relative P-Rex1/ β -Actin”. We have made similar amendments throughout the rest of the manuscript where appropriate, eg Supplementary Figure S1G.

Figure 4C, loading controls in S4 for all of the blots added to the figure, not just the ER loading control. The loading control for the P-Rex1 blot suggests that P-Rex1 may not be upregulated as the authors state in the text.

The corresponding loading controls for each blot have now been included in the main figure. We have performed densitometry of P-Rex1 compared to the matched loading control in each case. The densitometry from this analysis is shown in Figure 4D and shows that P-Rex1 is significantly upregulated.

Figure 4D, which loading control was the P-Rex1 compared to?

The P-Rex1 protein expression was compared to the matched β -Actin loading control from the P-Rex1 western blots in each case. The figure has been amended so that the y-axis now states “Relative P-Rex1/ β -Actin”.

Based on the loading controls of the Pak2 blot in S4, there may be an upregulation of Pak2 after Tam treatment, in contrast to the authors statement.

Thankyou to the reviewer for this observation. We have now performed densitometry, and there is a statistical difference in Pak2 protein expression between PyMT and PyMT-Tam cell lines. The quantitation data has been added as Supplementary Figure 5C, and the text has been altered at lines 479-480.

Reviewer #2 (Remarks to the Author):

Summary: In this manuscript, the authors identified a novel prognostic biomarker predicting late recurrence in estrogen receptor (ER)-positive breast cancer. They first sorted endocrine therapy resistant MCF-7 ER+ breast cancer cells into a fast and slow growing population and characterized their cellular characteristics and metastatic potential using in vivo xenograft models. Using single-cell RNA sequencing (scRNA-seq) analysis, they found that RAC1 signaling is upregulated in the slow-growing population compared to the fast-growing population. In addition, they showed that the expression of P-Rex1 (RAC1 signaling component) is increased in ER+ metastases and is associated with late recurrence.

The present manuscript is potentially important because it proposes a novel prognostic biomarker and therapeutic target for late recurrence in ER+ breast cancer. However, there are several major points that should be addressed to substantiate the authors' claims.

Major points:

1. Figure 2A: The authors concluded that resistant populations do not arise from pre-existing drug-tolerant persistent (DTP) cells due to the lack of obvious overlap between the parent and resistant cells. However, this approach is too ambiguous and subjective to support this argument. In addition, we cannot exclude the possibility that DTP cells are induced and activate a transcriptional program that confers a selective advantage upon drug treatment. To rigorously address this issue, the authors should use a cellular barcoding system to track the lineage of cells over time.

We thank the reviewer for this comment. The experimental model took us about 3 years to develop (the cells took >2 years to derive in vitro, and then 10 months of in vivo experimentation to validate). Our initial study was exploratory as we didn't know if these cells would grow differently in vivo. Repeating the experiment with barcoding would be extremely valuable and provide more definitive evidence about DTPs as rightly pointed out by the reviewer. However, the timeframe to undertake this is beyond the scope of the manuscript review period. In addition, the discussion of persister cells is a secondary observation of the manuscript rather than our major finding. We believe our major observation is the identification of elevated Rac signalling within slow growing cells that could be a driver in recurrence.

The reviewer's point is now explicitly acknowledged within the manuscript and we state:

Lines 316-323: "Endocrine tolerant cells did not bear a resemblance to one particular subgroup of parental cells (Figure 2A), in line with a prior study on a model of ER+ aromatase therapy resistance which suggests that endocrine therapy resistance does not arise from particular subclones (31). This does not eliminate the possibility of a persister cell population as we may not have examined sufficient cells, or used the most sensitive method to detect persister cells. However, without a strong indication of persister cells, we hypothesised that differences in proliferation during resistance could alternatively arise from heterogeneity between individual cells in their cell cycle response to endocrine therapy."

In the discussion:

Lines 712-714: "We also do not see strong evidence that the cells arise from a precursor population in the parental cells, although a barcoding approach would be needed to show this definitively."

2. Supplementary Figure 2A and 2B: The estimated cell cycle phase by DNA content (Supplementary Figure 2A) and scRNA-seq (Supplementary Figure 2B) is not consistent. This issue should be carefully investigated, as the main results from scRNA-seq (Figure 2) are highly dependent on the cell cycle phase predictions.

We agree with reviewers #1 and #2 that these cell cycle proportions are not superimposable, and this raises a question about the underlying biology identified in the scRNAseq.

Our original analysis used the CycleBase gene database to annotate the cell cycle. Since performing this analysis other gene lists have become available and used routinely to perform cell cycle annotation of scRNAseq data. Consequently, we re-analysed the data using (<https://satijalab.org/seurat/reference/cc.genes.updated.2019>) to determine if this was the source of difference between estimated cell cycle phases. While the data generated from this analysis is not identical to our original analysis, the new analysis delivered a very similar result, now shown in amended Figure 2D.

After regenerating these data, we considered carefully what may be the origin of the discrepancy in the cell cycle profiles generated by scRNAseq and by propidium iodide staining. We believe the discrepancy is due to the unique nature of fulvestrant arrest and estrogen stimulation within the experiment.

In this experiment we first arrest the cells with fulvestrant, which synchronises the cells in an arrest state. The addition of estrogen then stimulates a rapid induction of cell cycle genes (within 6h) that does not translate into a change in DNA content until >20 hours later. For example, PCNA and CCNE2 are part of the S phase transcriptional cell cycle, and these transcripts are known to become elevated within 6h of estrogen stimulation, with protein levels increasing shortly thereafter. However, the 3N DNA content of the cells (ie partially replicated DNA) is not readily apparent by propidium iodide staining until about 20h post treatment. We have documented this in our previous publications (1, 2). Figure 1 from (1) is reproduced here (Figure R2), showing that CCNE2 transcripts (Panel E of Figure R2) are elevated 6h post fulvestrant arrest and release into the cell cycle, but that the 3N DNA content is not readily apparent until at least 20h post estrogen stimulation (Panel A of Figure R2). In Musgrove et al (2), the following transcripts are significantly elevated within 6h of estrogen stimulation following fulvestrant arrest: ATAD2, CCNE2, CDC6, DTL, FEN1, MCM4, MCM6, MCM7, PCNA, POLR1B, RFC2, TYMS. These 12 genes all occur in the 43 gene signature (<https://satijalab.org/seurat/reference/cc.genes.updated.2019>) that is routinely used to annotate scRNAseq data to identify cells in S phase. Consequently, a 6h estrogen treatment of fulvestrant synchronised cells will give a signature of transcriptional S phase, but DNA content will not be observed to change until much later.

PUBLISHED DATA FIGURE REDACTED

Figure R2: Reproduction of Figure 1 from Caldon, C.E., et al., Estrogen regulation of cyclin E2 requires cyclin D1, but not c-Myc. *Molecular and Cellular Biology*, 2009. 29(17): p. 4623-39. This figure shows the induction of cell cycle genes and proteins following arrest with fulvestrant and stimulation with estrogen.

From this analysis we believe that, while transcriptional cell cycle signatures can normally be good surrogates for cell cycle position by DNA content, this may not be the case in the analysis of cells undergoing synchronised cell cycle arrest with fulvestrant followed by induced cell cycle entry with estrogen. It is possible that other cell cycle synchronisation models could also be difficult to interpret using these transcriptional cell cycle signatures.

Based on these comments and those of reviewer #1 we have amended the text so that these points are clear, including:

- *we refer to the previous papers that describe this model system and the timing of changes to DNA content as measured by propidium iodide staining;*
- *we differentiate DNA content analysis by propidium iodide staining from cell cycle prediction by scRNAseq, and don't use these terms interchangeably;*
- *we have rewritten the text and figures to not refer to cells being in a particular cell cycle phase, but to instead be in states of "fulvestrant arrest" or "estrogen stimulation".*

We believe these amendments now account for the discrepancy between the scRNAseq prediction and the cell cycle state as measured by propidium iodide. In addition, we believe that the inclusion of these data could now be useful for other researchers should they encounter discrepancies between cell cycle prediction and measured DNA content in synchronisation models.

Thankyou to both Reviewers for this input, as this has clarified the manuscript.

(1) Caldon, C.E., et al., *Estrogen regulation of cyclin E2 requires cyclin D1, but not c-Myc*. *Molecular and Cellular Biology*, 2009. 29(17): p. 4623-39.

(2) Musgrove, E.A., et al., *Identification of functional networks of estrogen- and c-Myc-responsive genes and their relationship to response to tamoxifen therapy in breast cancer*. *PLoS ONE*, 2008. 3(8): p. e2987.

3. Figure 2D and 2E: In the pseudotime analysis, it is unclear what the biological meanings of the inferred pseudotime are in this context. How robust are the inferred trajectories across replicates and different inference algorithms?

Our initial analysis, conducted in 2019 using an early version of the Seurat package (v2.3.4), has been updated to address the reviewer's concerns about robustness. We have re-analyzed the data using the latest Seurat version (5.1.0), which incorporates improved pipelines. This re-analysis yielded similar plots and cell clustering, with only minor alterations in enriched genes and cluster cell numbers. Importantly, our major findings remain consistent.

Regarding the reviewer's inquiry about additional replicates and different inference algorithms, we acknowledge the validity of this point. However, as mentioned in response to Q1, generating further biological replicates would require several years. It is crucial to note that our cell line models were intended as a discovery dataset, which successfully led us to identify the significance of the Rac pathway in late recurring breast cancer - the primary finding of our manuscript.

We also recognize that alternative inference algorithms might produce different trajectories. Our focus, however, was not on speculating about the specific ordering of cells along the trajectory. Instead, we concentrated on the fate of fulvestrant-treated cells. A key outcome of our trajectory analysis was the identification of two distinct fulvestrant treatment nodes, with one showing the greatest overlap with fulvestrant resistance. The signatures we identified within this overlapping node suggested a senescence-like phenotype, which we subsequently validated using orthogonal β -galactosidase assays. Thus, while we acknowledge the potential for different results with other algorithms, we believe our chosen pseudotime analysis proved relevant and valuable, as evidenced by our ability to orthogonally validate the findings.

We have carefully reviewed the text to ensure we are not overinterpreting the trajectory analysis. We have rewritten the text as follows:

Lines 337-350: "We next examined how fulvestrant arrested and estrogen stimulated cells transition through different states by plotting cells on a transcriptional trajectory using pseudotime analysis (17). This identified cells transitioning between six different nodes (Figure 2D). Estrogen treatment for 6h superimposed with four of the nodes, and these were enriched with S phase transcriptional signatures. Estrogen treatment for 24h localised across several nodes, but notably co-occurred with an area of G₂M transcriptional signature enrichment that was situated between nodes (Figure 2E). Fulvestrant treatment associated predominantly with G1 transcriptional signatures, which co-localised with the remaining two nodes (labelled Node #1 and Node #2). Node#1 and Node #2 were notably distant from one another on the x_pseudo axis (Figure 2D/E), which suggested that the two nodes enriched for Fulvestrant treatment could be distinct in terms of their transcriptional programming. We observed that the two resistant cell lines, Fast and Endocrine Tolerant, were co-located with Node #2 of fulvestrant treatment (Figure 2D/E), so we investigated the transcriptional signatures of the fulvestrant treatment enriched nodes in more detail to understand their differences."

4. Figure 2G: The upregulation of MT-ATP8 may be explained by the increased percentage of reads mapped to mitochondrial genes, a key technical feature of low-quality cells.

The reviewer makes a valid point. In our original analysis we included a cut-off for cells with high mitochondrial gene content in order to select out low quality cells, but this may not have been sufficient, and high mitochondrial gene expression could still have been an indicator of low quality. For clarity we have now selected a different gene to represent the induction of a senescent phenotype, and used loss the gene HMGB2 as an indicator of senescence (1).

The text at lines 357-361 now reads:

“We identified that HMGB2, whose reduction is a key indicator of senescence (34) is specifically reduced in Node #2 cells compared to other populations (-2.10 logFC, Figure 2G). This indicated that Node #2 could be a pseudo-senescent population as senescent cells frequently skip mitosis upon cell cycle (35), have sustained mitochondrial function (36), and have reduced HMGB2 (34).”

(1) Aird, K.M., et al., HMGB2 orchestrates the chromatin landscape of senescence-associated secretory phenotype gene loci. J Cell Biol, 2016. 215(3): p. 325-334.

5. Figure 2I and 2L: Line 318-320 on page 15 is unclear. What does “its substantial clustering with Arrest #2” mean?

We apologise to the reviewer for our original wording. We have now rephrased this section to improve clarity.

The beta-galactosidase level of the “Fast growing resistant” population was more similar to that of “48h Fulv” compared to the “Endocrine tolerant” population in Figure 2L, which is not consistent with Figure 2I?

Our original images were at high magnification and did not show sufficient cells to give a good representation of the results of the experiment. In order to ensure the robustness of the data, we have repeated the beta-galactosidase staining experiment, and provided new images and quantitation data. We also note that Figure 2I (now figure 2H) only shows the subset of cells in each population that occur within Node #2, whereas the beta-galactosidase staining in Figure 2J/K represents the entire population, so these analyses cannot be directly compared.

6. Figure 6F-H: The authors showed that RAC1 inhibitors reduce the RAC1 activity in vivo. However, this is not surprising. To demonstrate the efficacy of the RAC1 inhibitors, they should show that these drugs have a synergistic effect on reducing cancer cell survival and drug resistance with endocrine therapies in vivo.

Thankyou to the reviewer for this point.

We have now completed a patient derived xenograft model study of ER+ breast cancer treated with Rac inhibitors + endocrine therapy. We pre-screened a number of ER+ PDX models for those which expressed high P-Rex1, and selected the KCC-4653 model for further study (Supplementary Figure

8A). This model was derived from a patient whose cancer was ER+, and had progressed on multiple endocrine therapies (Figure 7A).

We then performed a study comparing single arm Rac inhibition (NSC23766 or R-ketorolac) with vehicle, and also comparing the combination of each Rac inhibitor with tamoxifen to tamoxifen alone (Figure 7B). We observed that Rac inhibition by R-ketorolac reduced tumour burden (Figure 7C), and that either Rac inhibitor significantly prolonged survival compared to vehicle treated mice (Figure 7D). Tamoxifen as a single agent was also able to reduce tumour burden and prolong survival, but the PDX continued to grow despite tamoxifen treatment, which was consistent with progression in the patient. Most importantly, co-treatment with tamoxifen and either Rac inhibitor led to tumour regression, and significantly less tumour burden than those mice treated with tamoxifen alone (Figure 7C, 7E).

Minor points:

1. “Figure 1O” at line 256 on page 12 and “Fig 1P” at line 259 on page 12 should be replaced with “Figure 1N” and “Figure 1O”, respectively.

Apologies for these errors, these figure numbers have been replaced.

2. Figure 2G, 3I: The statistical significance should be provided.

The data in Figures 2G and 3I has been re-extracted following the re-analysis of all scRNAseq data. Statistical comparisons have now been inserted between the different samples in Figure 2G and in Figure 3H (new version of Figure 3I).

3. Figure 2J: The resolution of the heatmap should be improved. What are the common upregulated genes between “48h fulv” and “Tolerant” and their shared biological characteristics?

In our new analysis the genelist has been selected to show a smaller number of genes so that they can be viewed at higher resolution (now Figure 2I).

We have inserted text identifying the top common genes and describing their characteristics on lines 368-374 of the revised manuscript:

“Top genes upregulated in both endocrine tolerant and fulvestrant arrested cells (PREX1, CPE, NPNT, KY, MALAT1) included several associated with metastatic and growth promoting functions in cancer. PREX1 is a component of the Rac signalling pathway that is associated with breast cancer metastasis (37), CPE suppresses apoptosis to promote tumour growth and metastasis in multiple cancer types (38) NPNT encodes nephronectin, an extracellular matrix protein that enhances bone metastasis (39), and the lncRNA MALAT1 has been associated with both the activation and suppression of breast cancer metastasis (40, 41).”

4. Many supplementary figures (e.g. Supplementary Figure 2C and 3H) were not referenced in the main text, or were referenced inaccurately, and were unedited.

We apologise to the reviewers for our poor annotation of the supplementary figures and poor referencing. All supplementary figures have been reviewed and edited to provide clearer labelling, and the cross-referencing in the main text has been proof read. Specific queries regarding individual supplementary figures have also been addressed below in the answers to 6 and 7.

5. Figure 3B and 3C: What do the node colors mean?

Due to the re-analysis of the scRNAseq datasets, this figure has been amended in the revised manuscript. In the original manuscript the colour gradient reflected the size of nodes. However, in the amended manuscript we have changed the presentation to match the data types in the analysis. The size of the node represents the degree of interconnectedness or number of connections between each node, with nodes with >8 encircled in bold. Brown nodes signify genes, light blue is phenotype, dark blue is a protein complex, pink is a protein family, as per SIGNOR signaling network analysis. This is now clearly stated in the figure legend.

6. “Supp Fig S3H” at line 374 on page 17 should read “Supplementary Figure 3I”. Then, Supplementary Figure 3H is not referenced.

We have corrected the text so that Supplementary Figures 3H and 3I (now 4H/4I) are correctly referenced within the main document.

7. Supplementary Figure 4: No labels. Supplementary Figure 4A referenced on line 400-401 on page 18 does not exist.

An old version of supplementary figure 4 that lacked labels was inadvertently included with the original submission. This has now been corrected in the replacement figure, Supplementary Figure 5, in the figure legend for Supplementary figure 5, and in the main text.

8. Figure 4I: The signal activity should be quantitatively compared.

In Figure 4I we have seen focal areas of intense signal. Identifying the boundaries of these areas is subjective, and for this reason we did not want to include quantitation. The descriptive and observational nature of these data has been made clearer by rewording:

“We observed that Rac1 activity was not uniform across the tissue, but instead there was heterogeneity, with regions of high and low activity within tumours chronically treated with tamoxifen (Figure 4I).”

9. Figure 5E and Supplementary Figure 5C: In the axis labels, “tSCNE” should be replaced with “t-SNE”.

These labels have been replaced.

Reviewer #3 (Remarks to the Author):

This work investigates the properties of slow proliferating ER+ breast cancer cells that can withstand hormone therapies. The authors observe that these have many different properties from fast growing resistant cells, notably low EGFR signaling and high Rac1 activity, possibly linked to Prex1. Interesting, they are at least as efficient as fast-growing cells in their ability to metastasize. Through a neat series of experiments and analysis, the authors demonstrate that hormone therapy selects for high Rac1 activity in the MMTV-PyMT model and potentially in patients. They conclude by showing the potential utility of targeting Rac1 pharmacologically. This is done using the widely used NSC23766 compound and an enantiomer of the analgesic ketorolac. Intriguingly, analysis is presented to suggest that ketorolac has anti-metastatic properties not observed in other NSAID analgesics. There is much to like about this work; it is interesting, novel, well-written, and broadly convincing study. However, there are some notable gaps and loose ends that need addressing before publication.

1. There is an interesting change in the collagen network in the slow tolerant MCF7 cells (Fig1j). Is this observed in the PyMT model – SHG images should be shown? Does the transcriptomic data provide possible explanation for the molecular basis? Is Rac1 activity driving the altered collagen (this can be determined in experiments suggested below)? Do the tumours have more CAFs – α SMA staining or similar would be informative?

Thankyou for this question and we have addressed each point individually below:

SHG images for PyMT model: *The PyMT model of slow growing cells (shown in Figure 4B-F) was derived in vitro and we are unable to perform SHG studies on in vitro cells. The in vivo model of PyMT (Figure 4G-I) was not of selected cells, but instead we examined spontaneous tumours for the occurrence of Rac high subpopulations. Thus, we are unable to perform analysis to correlate SHG output with slow growing cells within this in vivo PyMT model.*

Transcriptomic data: *The reviewer asks an interesting question about the transcriptome data and whether it suggests a molecular basis for changes in the collagen network. As the transcriptome data was collected from in vitro models, we are lacking the complexity and stimulation of the in vivo environment in those data. Further studies which perform transcriptome studies on the tumours would be valuable to address this point, and would be an area to further develop this project in the future.*

Does Rac1 activity drive altered collagen: *Thankyou to the reviewer for this question, and this is discussed under point 5, as suggested.*

Do the tumours have more CAFs: *We have performed α -SMA staining to determine if the tumours have more CAFs, and whether this could be influencing the deposition of collagen. We stained primary tumours associated with Figure 1I of the revised manuscript with α -SMA to identify fibroblasts, and quantified the α -SMA signal using percentage of cells positive.*

The data is shown in revised Supplementary figure 1H/I. In essence, we did not find that the level of CAFs (as indicated by α -SMA IHC) differed between the fast growing and the slow growing tumours. Interestingly we observed that the α -SMA staining patterns indicated possible different organisation of the fibroblasts when comparing the tumours from slow growing and fast growing cells. In tumours from fast growing cells the α -SMA staining formed thin parallel tracks through the tissue, whereas the α -SMA staining in tumours from slow growing cells formed more circular patterns with larger localised regions of staining, although not more staining overall. It could be that increased collagen

deposition in tumours from endocrine tolerant cells is related to these larger regions of CAF localisation.

As the data is observational, and difficult to quantitate, we have not included any speculation that these denser regions are the origin of the increased collagen within the revised manuscript. We have instead stated at lines 291-293.

“We did not observe any significant numerical differences in cancer-associated fibroblasts between tumours, as measured by α -SMA immunohistochemistry (Supplementary Figure S1H/I).”

Overall, we thank the reviewer for this interesting point, and this will become an endpoint in our continuing studies in this area to determine if CAF proportion and organisation could be important in slow growing tumours.

2. Do the authors observe similar fast and slow populations when selecting the resistant PyMT cells? This analysis is required to connect different parts of the paper.

We did observe fast and slowing cells when selecting resistant PyMT cells. Representative images have now been added to Supplementary Figure 5A and the following changes made to the text:

Lines 473-477: “MMTV-PyMT Rac1-FRET cells were chronically exposed to tamoxifen for ~30 passages until reaching a constant proliferative rate (PyMT-Tam cells). Treated cells initially showed significant heterogeneity of colony size, indicating the presence of fast and slow growing colonies (Supplementary Figure 5A).”

3. Why are the Prex1 high and low groups asymmetric in the Kaplan Meier analysis (Fig 5j). They should really be split into equivalent size high and low groups or tertiles. The asymmetric split suggests some selective choosing of an optimal cut-off. Can the authors either justify their cut-off or re-plot the analysis as suggested?

Our original analysis used an H-score cutoff of 50. An H-score cutoff of <50 is commonly considered as negligible expression for many protein biomarkers, and frequently used in analysing breast cancer samples (1,2). For this reason, we prepared Supplementary Figure 6E to see if this represented a natural cut-off in the data, and we saw that H-score of 50 represented a less dense area of the dot plot among all the samples that were stained, and was a suitable cut-off.

Following on from this comment we have re-examined all cut-offs throughout the manuscript. The original cut-offs in Figures 5G and Supplementary Fig 6D were derived from an automated “find best cut-off” algorithm as described in (3). For consistency we have now performed the analysis on equal groups for all genes. Consequently, we have reanalysed the data for PREX1 and MKI67 in Figure 5, and RAC1, PAK2 and EGFR in Supplementary Figure 6.

Our analysis with symmetrical cut-offs shows the same relationships, although the statistical significance has changed. In the analysis of long term recurrence and its relationship with PREX1 expression, the survival curve is now $P < 0.0748$, which does not reach the statistical significance cutoff of $P < 0.05$. The hazard ratio remains low at 0.36. We found that the significance values and hazard ratios remained very similar for MKI67, RAC1, PAK2 and EGFR associations with progression free survival following endocrine therapy.

While the data for PREX1 association now show a p-value of 0.0748 with late recurrence, we believe that contrast of the shape of the survival curves between PREX1 and MKI67 still remains important, especially in the context of the other data presented in Figure 5. Each dataset shows an inverse relationship between PREX1 and proliferative markers, and that high PREX1 occurs in late recurring breast cancers, and is not associated with good prognosis in that setting. As we no longer see a relationship that reaches the cut-off of $P < 0.05$, we have moderated the description of our findings.

In the text we now state:

“Within the first 5 years of endocrine therapy, there was no difference in recurrence-free survival of patients with high/low PREX1 expression, however recurrence-free survival was lower in patients with high PREX1 after 5 years ($p < 0.075$; Figure 5G). High expression of RAC1 or PAK2 showed an association with poor recurrence-free survival throughout recurrence (Supplementary Figure 6D).”

- (1) Jeon T, Kim A, Kim C. Automated immunohistochemical assessment ability to evaluate estrogen and progesterone receptor status compared with quantitative reverse transcription-polymerase chain reaction in breast carcinoma patients. *J Pathol Transl Med.* 2021 Jan;55(1):33-42. doi: 10.4132/jptm.2020.09.29. Epub 2020 Dec 3. PMID: 33260290; PMCID: PMC7829576.
- (2) Leone JP, Bhargava R, Theisen BK, Hamilton RL, Lee AV, Brufsky AM. Expression of high affinity folate receptor in breast cancer brain metastasis. *Oncotarget.* 2015 Oct 6;6(30):30327-33. doi: 10.18632/oncotarget.4639. PMID: 26160847; PMCID: PMC4745802.
- (3) Lánczky A, Gyórfy B. Web-Based Survival Analysis Tool Tailored for Medical Research (KMplot): Development and Implementation. *J Med Internet Res.* 2021 Jul 26;23(7):e27633. doi: 10.2196/27633. PMID: 34309564; PMCID: PMC8367126.

4. The authors should attempt staining therapy resistant patient samples with a Rac1-GTP antibody and EGFR (or another fast resistant marker).

Thank you to the reviewer for this suggestion, which we address for each staining approach separately below.

Rac1-GTP staining: While we would like to address the point of the reviewer, we don't believe that it is technically possible with available reagents. This is because there is concern in the Rac field about the utility of antibodies to Rac1-GTP, where publications report on the high affinity of these antibodies for vimentin instead of Rac1-GTP, and how this has led to misleading results (1,2). For this reason, we have adopted a biosensor approach for most of the Rac activity studies described in our manuscript, and this has been extensively validated by us previously (3). Unfortunately, the biosensor approach is not suitable for analysis of patient samples, and GEFs, such as P-Rex1, are often used as surrogate markers for Rac activity in patient cohorts.

- (1) Baker MJ, Kazanietz MG. The anti-Rac1-GTP antibody and the detection of active Rac1: a tool with a fundamental flaw. *Small GTPases.* 2022 Jan;13(1):136-140. doi: 10.1080/21541248.2021.1920824. Epub 2021 Apr 29. PMID: 33910489; PMCID: PMC9707529.
- (2) Baker MJ, Cooke M, Kreider-Letterman G, Garcia-Mata R, Janmey PA, Kazanietz MG. Evaluation of active Rac1 levels in cancer cells: A case of misleading conclusions from immunofluorescence analysis. *J Biol Chem.* 2020 Oct 2;295(40):13698-13710. doi: 10.1074/jbc.RA120.013919. Epub 2020 Aug 14. PMID: 32817335; PMCID: PMC7535912.

- (3) Floerchinger A, Murphy KJ, Latham SL, Warren SC, McCulloch AT, Lee YK, Stoehr J, Méléneć P, Guaman CS, Metcalf XL, Lee V, Zaratian A, Da Silva A, Tayao M, Rolo S, Phimmachanh M, Sultani G, McDonald L, Mason SM, Ferrari N, Ooms LM, Johnsson AE, Spence HJ, Olson MF, Machesky LM, Sansom OJ, Morton JP, Mitchell CA, Samuel MS, Croucher DR, Welch HCE, Blyth K, Caldon CE, Herrmann D, Anderson KI, Timpson P, Nobis M. Optimizing metastatic-cascade-dependent Rac1 targeting in breast cancer: Guidance using optical window intravital FRET imaging. *Cell Rep.* 2021 Sep 14;36(11):109689. doi: 10.1016/j.celrep.2021.109689. PMID: 34525350.

Proliferative marker staining: To understand the proliferative rate in the therapy resistant samples we have used KI67 as a proliferative marker, as this has been used throughout the rest of Figure 5. We did not observe a difference in P-Rex1 staining between primary tumours and metastases based on whether the tumours increased or decreased in staining intensity for KI67 (Figure R3 below).

While analysing these data, we noted that all samples generally had very low KI67 in both primary and metastatic disease, with most samples showing KI67 in less than 10% of cells. This means that large differences in proliferation were not detected across the cohort, which was consistent with this being a cohort of slow growing, late recurring, cancers. While we see a negative correlation between P-Rex1 and Ki67 in other cohorts in Figure 5, we note that these are generally mixed cohorts that include triple negative breast cancers (Figure 5A-D), and both early recurring and late recurring ER+ disease (Figure 5G). Thus, a negative correlation may be more readily detectable in those settings as they include both fast growing and slow growing cancers. This is in contrast to the cohort shown in Figure 5K-5L. This cohort was selected on the basis of peritoneal recurrence, which occurs at higher frequency in patients with late recurring breast cancers.

There was also a lower rate of staining success from the samples than in the analysis of P-Rex1 versus ER expression in Figure 5K-5L. This analysis would benefit from a larger number of samples in order to have sufficient power to see any signal.

As a result, the data have not been included in the main manuscript, as we are unsure if the lack of difference that we observe is due to (a) the low intensity of staining, (b) the small sample size, (c) the naturally low proliferation rate across the cohort, or (d) whether it truly represents a null result. We have included the data here in the response document (Figure R3).

UNPUBLISHED DATA FIGURE REDACTED

5. The Rac1 dependence of the slow tolerant cells is never demonstrated in vivo. This is critical and needs to be shown. Ideally, by long-term treatment with NSC23766 or ketorolac, or shRNA/Crispr KO of Rac1. These tumors should be analyzed for collagen SHG (c.f. point #1)

In Figure 6 we demonstrated the loss of Rac signalling in tamoxifen treated tumours when mice are co-treated with NSC23766 or R-ketorolac. Unfortunately, these models are not suitable for long-term treatment, as window studies can only be carried out for a maximum of 4 days. We considered performing a therapy experiment in the model presented in Figure 1 which takes 9-12 months to take to completion. A major limitation in using this model is that our animal ethics limits the administration of drugs by oral gavage to a period of 90 days (4.5 months administered 5 days/week), so we would be unable to administer Rac inhibitors throughout the experiment.

Thus, to address this question and also the question of Reviewer #2, we screened ER+ PDX models for those high in P-Rex1 and Rac signalling in order to perform an in vivo study on a clinically relevant model. The 4653 PDX model was derived from a patient who had progressed on several endocrine therapies (tamoxifen, letrozole, exemestane) as well as chemotherapies and CDK4/6 inhibitors. The tumours are partially responsive to tamoxifen in vivo, but eventually the mice reach ethical endpoint. This is a slow growing PDX model that takes ~ 4 months to reach endpoint without therapy.

We treated this model with vehicle, Rac inhibitor NSC23766, Rac inhibitor R-ketorolac, tamoxifen, and tamoxifen combined with either Rac inhibitor. The model shows a partial response to either Rac inhibitor as single agents, leading to a significant decrease in tumour volume with R-ketorolac treatment, and a significant increase in survival with NSC23766 or R-ketorolac treatment (Figure 7C). Tamoxifen, as expected, is quite efficacious in this model, but does not elicit a lasting response. By contrast, the combination of tamoxifen and either Rac inhibitor led to tumour regression (Figure 7C/E).

We assessed the endpoint tumours of the six treatment arms for collagen. For vehicle, NSC23766 and R-ketorolac this occurred at the ethical endpoint of tumour volume. For tamoxifen, tamoxifen + NSC23766 and tamoxifen + R-ketorolac this occurred at the timed endpoint of 18 weeks of treatment (corresponding to the maximum amount of drug that could be administered under our animal

ethics). In these sets of tumours we used Picrosirius Red to assess collagen deposition so that we could take advantage of polarised light microscopy to distinguish different types of collagen.

In the longer term samples (18 weeks: tamoxifen, tamoxifen + NSC23766, tamoxifen + ketorolac), we saw that all tumours have higher intensity signal for picrosirius red than vehicle treatment, which could be a function of time, or a function of tamoxifen treatment. Surprisingly, the Rac inhibitor treated tumours had thicker collagen tracks than those treated with tamoxifen alone, although this was not significant (Figure 7F/G).

Based on these data we did not observe a reduction in collagen following Rac inhibitor treatment, however there may be remodelling occurring in response to Rac inhibition. This is an interesting area of future investigation for the lab.

6. Is Prex1 actually required for the growth and metastasis of the slow tolerant cells? Or is it that they are simply Rac1 dependent. The former point needs to be tested experimentally using shRNA/Crispr KO.

Thankyou for this point, we have now experimentally tested this question using PREX1 shRNA, and examined the effect of PREX1 shRNA on the migration of fast growing and endocrine tolerant cells. PREX1 shRNA derivatives of endocrine tolerant cells showed a ~37% impairment in the migration of cells, whereas PREX1 shRNA derivatives of fast growing cells did not show a significant difference in migration potential compared to a non- targeting control. These data are now shown in Figure 3P/Q.

The text has been altered:

Lines 438-446: "To determine whether P-Rex1 was important to the oncogenic potential of tolerant cells, we performed PREX1 shRNA experiments and examined the effect on the migratory ability of cells, which is an important facet of P-Rex1 activity in cancer (49). PREX1 shRNA reduced expression of P-Rex1 in both the Endocrine Tolerant and Fast growing resistant cells (Figure 3P, Supplementary Figure 4I). Endocrine Tolerant cells expressing PREX1 shRNA showed a ~37% impairment in the migration of cells compared to a non-targeting control, whereas PREX1 shRNA derivatives of fast growing cells did not show a significant difference in migration potential compared to control (Figure 3Q). Thus, in Endocrine Tolerant cells the expression of P-Rex1 is important for migration."

Methods have been added in the supplementary data.

7. What are the cohort sizes in the analysis of ketorolac and other NSAIDs?

We apologise for not including this information in the original draft. We have now included a supplementary table with these information, Supplementary Table 6.

As Figure 6 was expanded to include the new PDX model, we note that the meta-analysis data are now included as a separate figure, Figure 8.

8. The authors don't discuss the topic of cancer dormancy very directly, but their study clearly relates to it. Some greater discussion is warranted and it would be interesting to cross-reference the transcriptome of the slow tolerant cells with signatures of dormancy.

The reviewer makes an excellent point, as within the manuscript we have not directly addressed where our models sit within the spectrum of concepts such as senescence, quiescence and dormancy.

We have found that Endocrine Tolerant cells show some signatures of senescence, including high β -galactosidase, and slow proliferation (though not cell cycle arrest). Prunier et al (2021) (1) have performed a study looking at models of dormancy, and found that dormancy and senescence were likely to be mutually exclusive states, suggesting that our model is unlikely to have characteristics of dormancy. However, we also compared our signatures to the transcriptome of dormant cells in the bone identified in a separate study of breast cancer cell dormancy (Ren et al (2022) (2)). Only a small proportion of these genes overlap, but interestingly Prex1 is one of the overlapping genes, perhaps indicating that dormant breast cancer cells engage Rac signalling.

We have added Supplementary Figure 3, which shows this comparison, and made alterations to the text as follows:

Results lines 395-398: “We additionally compared the Endocrine Tolerant cells to genes enriched in a dormancy model of ER+ breast cancer (42). This did not identify a large overlap of genes, and a similar proportion of overlap was seen between fast growing cells and the dormancy gene set (Supplementary Figure 3A/B).”

Discussion, lines 719-728: “The relationship between endocrine tolerant cells and previously characterized dormant cells also needs to be considered. One study found that dormant breast cancer cells lack senescence features (96), which is consistent with our observations of a senescent-like phenotype in slowly proliferating endocrine tolerant cells. We also compared the gene signature of endocrine tolerant cells with the gene expression profiles associated with an ER+ dormancy model (42), Supplementary Figure 3), but our analysis did not reveal significant similarities. However, it is noteworthy that PREX1 was identified as one of the 25 genes common to endocrine tolerant cells and the dormancy model. This overlap suggests that the Rac pathway may play a significant role in cells that have entered a more profound state of cell cycle arrest, such as that associated with dormancy.”

- (1) Prunier C, et al Breast cancer dormancy is associated with a 4NG1 state and not senescence. NPJ Breast Cancer. 2021 Oct 27;7(1):140.*
- (2) Ren, Q., et al., Gene expression predicts dormant metastatic breast cancer cell phenotype. Breast Cancer Res, 2022. 24(1): p. 10.*

RESPONSE TO REVIEWERS

REVIEWERS' COMMENTS

Reviewer #1 (Remarks to the Author):

The authors have made many modifications of the manuscript in response to reviewer comments. In addition, they have also added important experimental support for their hypothesis, all of which have materially improved this study. There are still some additional concerns about this manuscript, but they require only text changes and edits, not additional experimental work

Thankyou to the reviewer for their valuable feedback, which has helped to improve the manuscript.

Concerns:

For figure 1M, the non-significant result for pulmonary metastases appears to be driven primarily by one sample in the endocrine tolerant group. While I applaud the authors for including all of the data, I wonder whether it is worth also analyzing the data with this animal censored. This shouldn't change the overall interpretation of the manuscript, but might better describe what is observed.

The reviewer makes an excellent point that this mouse may be an outlier. For transparency we will include all data in the manuscript. However, if conducting similar experiments in the future, we would elect to use a larger cohort of mice so that we can determine if events such as these are truly outliers.

Figure 1N, the authors state that the bone mets were "detectable in a subset" of samples. How were these subsets selected? Or are they just looking at those animals with detectable bone mets? Please clarify.

When this experiment was first designed the endpoint of the study was lung metastasis. However during the course of the experiment we suspected that metastases were occurring at other sites. The detection of bone metastasis was a new method for our research group, for which we had to receive training and buy the necessary reagents. The data shown in the figure represents all the samples for which we successfully performed a bone marrow extraction and performed staining for metastases. Other samples are not included as they were either not collected, or failed due to technical reasons as we were optimising the staining protocol. As we collected only a small subset of samples, including only two in one group, we did not perform statistical analysis on this data.

Figure 1O, the authors state that "all mice" showed evidence of gyn abnormalities, but the histogram shows only 8 of the 10 tolerant animals having these abnormalities. Please clarify.

All mice were scored for gynaecological metastases, but initially we were unsure of the appearance of two of the mice, and these were not included in the scoring. This was an oversight in preparing the manuscript, and to be rigorous we are now scoring those mice as negative. The data in Figure 1O has been adjusted to reflect this. This does not result in any changes to the conclusions of the manuscript.

Also, only 9 mice were shown for the fast growing cohort for pulmonary metastasis and gynaecological abnormalities. The remaining mouse died unexpectedly overnight and was found dead in its cage the next morning. Consequently, we were unable to perform a necropsy for this mouse and it is not included in the data for metastatic burden.

For transparency, we have now included the above details on the number of animals in sub-studies in the "Data Exclusions" section of the Reporting Summary.

Are the number of epithelial cells sequenced in scRNA seq provided somewhere? I may have missed this but it would be worthwhile reporting

These numbers have been inserted in the methods section.

Figures 3L, the Y-axis state normalization to b-actin, but the blot in 3K, as well as the uncropped blot in the supplemental data (fig. S4D) is labelled GAPDH. The band in the uncropped blot also is at the appropriate size for GAPDH. Please correct the figure and associated text.

We apologise for this error and thank the reviewer for their careful reading of the manuscript. This error has now been corrected.

For the record, in shRNA experiments, it is usually better to analyze more than a single shRNA to avoid off-target effects. However, since the rest of the data is internally consistent, and considering the length of the review cycle for this paper, in this instance I am not suggesting any further delay to perform this additional work.

The point of the reviewer is well taken, and in extensions to this work we will try to identify further constructs that are effective at knocking down PREX1.

Please provide a y-axis label for figures 5I and 5J

This has been provided.

Figure 7C, for the legend, please move the tamoxifen only treatment to the Single-arm part of the legend rather than the Combination, for clarity.

The labels of single-arm and combination have been removed so that there is no confusion. This allows for the labels to be close enough together to provide statistical values.

Reviewer #2 (Remarks to the Author):

The authors have satisfactorily addressed my comments.

Thankyou to the reviewer for their valuable feedback, which has helped to improve the manuscript.

Reviewer #3 (Remarks to the Author):

The authors have made reasonable attempts to address the concerns raised or explained why they are not able to do so (e.g. Rac-GTP antibody issues). I recommend publication. My only query is the p value for Figure 5k - is there a typo, should it be <0.0149? If it is <0.149, then this is a rather odd way of indicating no significance.

Thankyou to the reviewer for their comments, and we are very please to have publication recommended. For clarity we have changed this value to n.s. to indicate not significant.

ADDITIONAL AMENDMENT TO FIGURE 8B

In Figure 8B of the revised manuscript we have made an amendment to remove two datasets. When we originally performed the meta-analysis, we split a cohort described in Forget et al 2010 (PMID: 20435950) into multiple subsets, and then included each subset in the meta-analysis in Figure 8B. Our first approach to the analysis was the same as was taken in a previous meta-analysis published analysing the same dataset (PMID: 34748112). However, we have since become aware that splitting the dataset in this manner can introduce a risk of duplicate inclusion bias into the analysis through multiple counting of the control subjects. For this reason, we have restricted the meta-analysis to one subset of data only from Forget et al 2010.

This change does not change the outcome of the analysis, and it does not change the conclusions of the manuscript. The different subsets of the study of Forget et al are still shown in Supplementary Table 6 to demonstrate that none of the drugs in the sub-studies are associated with improved recurrence free survival.

For transparency we wanted to highlight this alteration to the reviewers and editors.

This manuscript describes the investigation into endocrine therapy and late relapse for ER+ breast cancers. The authors utilize a drug selection method to isolate fast or slow growing cells resistant to endocrine therapy to try to mimic in an experimental setting the important phenomenon of late ER+ breast cancer relapse. Subsequently they perform a variety of in vivo, cellular and molecular analyses to investigate whether or not the slow growing, “Endocrine Tolerant” cells might be responsible for late metastatic relapse. They found that these cells were capable of forming primary tumors, but these tended grow slowly in vivo as in vitro, but had the same metastatic incidence as their fast growing complements. The authors then characterized these cells and determined that the Endocrine Tolerant cells had acquired some characteristics of senescent cells. Subsequently the authors performed scRNAseq to further investigate the molecular cause of these changes and implicated the Rac1 pathway molecule P-Rex1. Further investigations using a Rac1 biosensor and Rac1 pathway inhibitors suggest that this pathway may be important in metastatic progression in ER+ patients. If true, this has the potential of providing an avenue for eventual clinical intervention in this subset of patients which might significantly improve survival for the largest subtype of breast cancer.

Supplemental figure S1E, line 226, tolerant cells look like they are expressing less ER compared to fast cells when comparing relative ratio of ER to b-actin. Densitometry analysis would be helpful to quantitate the differences in ER between the two cell populations. This doesn't change the difference in growth rate between the two types of cells under fulvestrant or tamoxifen, but might explain the difference in growth in the presence of estradiol.

Figure 1M, are the average number of metastases observed in each sample significantly different between the fast and tolerant animals? Incidence and size, once formed, may not be significantly different, but the data suggests a significant reduction in the number of metastases in the slowly growing tumors. This would suggest a decrease efficiency, not an equal efficiency, as stated on line 263.

Line 273-4, figure 2A, I disagree. There is overlap between at least the endocrine resistant and the parental population in the center of the graph, which suggests that this population may have pre-existed selection.

Supplemental figure 2A & B, why are the cell cycle profiles so different? Determining cell cycle state by DNA content should be relatively accurate, with cells exhibiting 3N being considered in S phase. The pattern observed in S2A is consistent with expected re-entry to the cell cycle. Using a transcriptional profile for single cell sequencing, with inherent lack of sequence depth, introduces a potential significant source of error which would significantly affect the remaining data presented in figure 2D-F. The relative cell cycle percentages in S2A make more sense considering the treatments that these cells have undergone. If the subsequent analysis is done using the cell cycle assignments based on DNA content, rather than the transcriptional signature, are the same answers obtained or is the interpretation changed?

Figure 2K, are these images representative? This does not show a significant difference in b-gal staining. What criteria did the investigators use to call a cell as positive/negative?

Figure 3G, are the HER2 and pHER2 images from the same blot? The differences in the band patterns suggest that they are not. If this is correct, where is the b-actin control for the pHER2 blot?

Figure 3I, the y-axis is listed as “PREX1 relative transcripts”, relative to what? Same question for 3J.

GAPDH loading control for the top two panels in figure 3K is not included in the figure. Control is included in figure S3D. This should be included in the primary figure rather than just including the control for the ER blot (S3D right panels), as currently shown. The interpretation that p-Rex1 is up-regulated by estrogen supplementation is questionable. If it is true, it is very subtle. Densitometry analysis using the correct GAPDH loading control shown in S3D needs to be performed to substantiate this claim.

Supplemental figure 3C, the Rac1 blot appears to be from a different experiment than the others, based on the curved banding pattern in this blot compared to the others, specifically the faint lower band in the Pak2 blot. The b-actin control for this blot needs to be included in the supplemental figure and added made in figure 3J to reflect that this is from a different experiment.

Line 374, the METABRIC analysis is S3I, not S3H as written.

S3F and S3G are the duplicates

b-actin blot for figure 3N is not the same as the source blot in S3E. See superimposed image:

Figure 4D, what is the “Relative P-Rex1” expression normalized to?

Figure 4C, loading controls in S4 for all of the blots added to the figure, not just the ER loading control. The loading control for the P-Rex1 blot suggests that P-Rex1 may not be upregulated as the authors state in the text.

Figure 4D, which loading control was the P-Rex1 compared to?

Based on the loading controls of the Pak2 blot in S4, there may be an upregulation of Pak2 after Tam treatment, in contrast to the authors statement.